# Embodied Task Planning via Graph-Informed Action Generation with Large Language Models

Xiang Li[1 +]   Ning Yan[2]   Masood Mortazavi[2]

## Abstract

While Large Language Models (LLMs) have demonstrated strong zero-shot reasoning capabilities, their deployment as embodied agents still faces fundamental challenges in long-horizon planning. Unlike open-ended text generation, embodied agents must decompose high-level intents into actionable sub-goals while adhering to the constraints of a dynamic environment. Standard LLM planners frequently fail to maintain strategy coherence over extended horizons due to context window limitations or hallucinate state transitions that violate environment constraints. We propose GiG, a planning framework that structures embodied agents' memory using a Graph-in-Graph architecture. Our approach employs a Graph Neural Network (GNN) to encode environmental states into embeddings, organizing these embeddings into action-connected execution trace graphs within an experience memory bank. GiG enables retrieval of structurally-similar priors, allowing agents to ground current decisions in relevant past structural patterns. Furthermore, we introduce a bounded lookahead module that leverages symbolic transition logic to enhance the agent's planning capabilities through grounded action projections. We evaluate our framework on three embodied planning benchmarks—Robotouille Synchronous, Robotouille Asynchronous, and ALFWorld. Our method outperforms state-of-the-art baselines, achieving Pass@1 performance gains of up to 22% on Robotouille Synchronous, 37% on Asynchronous, and 15% on ALFWorld while maintaining comparable or lower computational cost.

+Work was completed during an internship at Futurewei Technologies. [1]Purdue University [2]Futurewei Technologies. Correspondence to: Xiang Li <li2068@purdue.edu>.

*Proceedings of the 43rd International Conference on Machine Learning*, Seoul, South Korea. PMLR 306, 2026. Copyright 2026 by the author(s).

## 1. Introduction

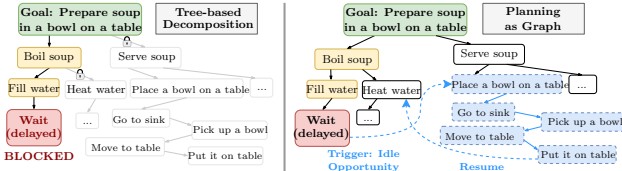

*Figure 1.* In tree-based decomposition (left), peer sub-goals are structurally blocked until the current node completes, forcing idle waits. In contrast, using a graph for planning (right) allows dynamic instantiation of new sub-goals, enabling the agent to interleave tasks and utilize the idle horizon.

Embodied task planning (Huang et al., 2022; Wu et al., 2023) refers to the ability of an embodied agent to perceive and interpret environmental observations, and to reason over these perceptions to generate coherent, multi-step action plans. These plans are grounded in the agent's interactions with the environment, enabling the agent to accomplish complex, long-horizon tasks that require sequential decision making, adaptation to dynamic environmental change, and coordination between perception, reasoning, and action. In embodied task planning, this process typically involves decomposing a high-level intent into a sequence of intermediate sub-goals, which are further decomposed or executed through interactions with the environment.

However, for sophisticated tasks, sub-goals are rarely independent; they exhibit intricate dependencies where early decisions heavily influence future feasibility. Consequently, an agent must perceive, adapt, and revise its strategy while maintaining a consistent reasoning chain over extended periods. Frameworks like ReAct (Yao et al., 2023b) and Reflexion (Shinn et al., 2023) interleave action and observation to incorporate environmental feedback into the generation loop. Although effective, these sequential, action-interleaved strategies face challenges in long-horizon settings as noted by ReCAP (Zhang et al., 2025b). The continually growing interaction history leads to context drift (Wu et al., 2025), where the limited context window causes models to lose track of high-level goals, resulting in repetitive or disjointed actions. To mitigate this, ReCAP introduced a hierarchical context tree (Figure 1 left) to maintain high-level goals, dynamically decompose sub-tasks generated by the

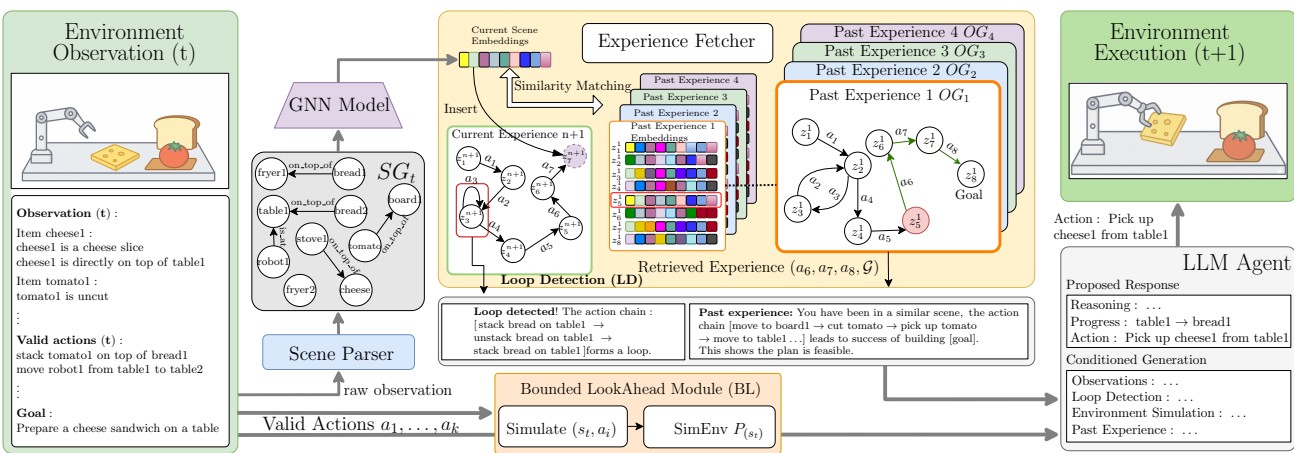

*Figure 2.* GiG parses the environment observation to build a scene graph, which is encoded by GNN as a structurally rich embedding. This embedding is fed into an experience fetcher to retrieve structurally similar past memory and detect exploration loops. An LLM agent generates the next action conditioned on current observation, past related experience, current goal, and bounded look-ahead results.

agent into atomic actions and perform backtracking upon failure. While this memory design is effective for top-down task decomposition, tree structures struggle to efficiently represent parallel sub-tasks, as they enforce an artificial serialization of concurrent events. For instance, in Figure 1, the "Fill water" task introduces delay (passive waiting). While actions such as "Place a bowl on a table" could occur simultaneously, a tree structure cannot model these interleaved events, as peer sub-tasks are structurally blocked until the current node is completed. Furthermore, a malformed high-level sub-goal can propagate errors downward, resulting in unnecessary exploration steps.

To this end, we propose GiG (Graph-in-Graph) (Figure 2), a novel framework designed for robust embodied planning. GiG maintains a two-tier topological memory: a local *scene graph* to capture immediate spatial relations, nested within a global *state-transition graph* connected by actions taken to track task progress and detect cyclic failures. The dynamically updated scene graph continuously aggregates new observations while refining existing ones, ensuring context-window efficiency and providing robustness against perception noise. The state-transition graph allows the agent to dynamically branch out and instantiate new sub-goals based on real-time feasibility, enabling the utilization of otherwise idle time, effectively breaking the "blocked sibling" constraint found in tree-based decomposition. Furthermore, GiG leverages a Graph Neural Network (GNN) to encode local scene graphs into structurally-aware embeddings stored in an experience memory bank, allowing the retrieval of historical execution patterns to guide similar future tasks. Additionally, we integrate a Bounded Lookahead (BL) module to equip the agent with proactive planning capabilities.

The key contributions of this paper are as follows.

- We introduce a novel Graph-in-Graph memory architecture, where a lightweight GNN encodes scene graphs into embeddings. Those embeddings form nodes which are connected by edges (actions) in an external state-transition graph, capturing environmental dynamics and facilitating experience retrieval to guide future exploration.

- We propose an optional Bounded Lookahead module to enable proactive optimization by grounding the agent's reasoning with environment constraints.

- We evaluate GiG on three embodied reasoning benchmarks Robotouille Synchronous, Asynchronous, and ALFWorld across LLM models of varying scales. GiG consistently outperforms state-of-the-art baselines while remaining computationally efficient.

## 2. Related Work

### 2.1. Structured Memory and Memory Architectures

Retrieval-Augmented Generation (RAG) (Lewis et al., 2020) is a widely adopted technique for equipping LLM agents with up-to-date, factual knowledge while avoiding costly fine-tuning. However, standard RAG retrieves isolated text chunks via similarity search, often failing to capture the intrinsic relationships between pieces of information (Zhu et al., 2025). Recent research (Procko & Ochoa, 2024; Edge et al., 2025) has demonstrated that retrieval quality can be significantly improved by using a structured Knowledge Graph (KG) instead of an unstructured text corpus. In these GraphRAG systems, agents leverage the graph's explicit structure to retrieve a semantically coherent subgraph of interconnected facts, rather than just a list of disconnected fragments. This principle has been adopted in several existing works. PoG (Chen et al., 2024) designs a self-correcting

framework over a static KG for Q&A tasks. HiRAG (Huang et al., 2025) utilizes a hierarchical KG to improve structural and semantic understanding. Beyond alternative approaches for structuring knowledge bases, recent literature also employs specialized memory architectures for complex LLM tasks. For example, EvoMem (Fan et al., 2025) introduces a dual-evolving memory mechanism to improve multi-agent coordination in planning tasks. Unlike existing approaches, our framework synthesizes two distinct graph structures into a two-tier memory: a spatial scene graph to capture environmental topology and a dynamic transition graph to explicitly model scene evolution over time. Our proposed approach treats structurally equivalent scenes as anchors to retrieve successful past execution traces, effectively using past experience as priors to guide the agent through similar environmental patterns.

## 2.2. Task Planning with LLM Agents

Prompting techniques like Chain-of-Thought (CoT) (Wei et al., 2022) and Tree-of-Thoughts (ToT) (Yao et al., 2023a) enhance reasoning in abstract domains. Graph-of-Thoughts (GoT) (Besta et al., 2024) introduces graph structures, but it remains an internal thought process rather than execution states. They are prone to hallucination and lack the grounding required for dynamic environments. Frameworks like ReAct (Yao et al., 2023b) mitigate this by interleaving action and environment observation to create a dynamic feedback loop, allowing the agent to continuously update its plans based on the feedback. Recent literature applies these principles to two primary planning domains: web navigation planning (Kim et al., 2024; Erdogan et al., 2025; Zhang et al., 2025a; Yang et al., 2025b; Cheng et al., 2025) and embodied planning (Song et al., 2023; Yoo et al., 2024; Liu et al., 2026; Gonzalez-Pumariega et al., 2025; Zhang et al., 2025b; Mon-Williams et al., 2025). We focus on embodied planning in this work, which operates in an interactive, physical environment (real or simulated), where the core challenges are reasoning about physical interactions, managing concurrent processes, and handling resource contention. LLM-Planner (Song et al., 2023) uses in-context example retrieval and replanning to improve sample efficiency and task completion rate. ExRAP (Yoo et al., 2024) uses a graph to keep the environment information fresh, which acts as a database for fact querying in the future. ExpeL (Zhao et al., 2024) summarizes insights from past experience and uses those insights to guide future tasks. ReCAP (Zhang et al., 2025b) uses recursive reasoning and decomposition to build a context tree for backtracking in robotics planning. Distinct from prior works, GiG adopts a graph-based memory that serves two purposes: (1) It enables flexible, non-blocking replanning; (2) It allows the agent to retrieve and transfer successful experiences from past episodes to guide execution in similar environments.

# 3. Proposed Method

## 3.1. GiG Memory Architecture

A primary challenge in long-horizon planning is transforming the unstructured observations (i.e., raw text) into a compact, relational, state representation (Zhang et al., 2025b). Flat text descriptions fail to capture the rich structural relationships between entities, and extended interaction histories lead to context drift (Wu et al., 2025). To overcome this deficiency, we introduce the Graph-in-Graph (GiG) memory architecture, which consists of a two-tier graph structure: a scene graph (inner graph) and a state-transition graph (outer graph).

### 3.1.1. SCENE GRAPH (INNER GRAPH)

We first transform the observations of the environment at step $t$ into a scene graph $SG_t = (V_t, E_t)$. While we employ a deterministic parser in our experiments to ensure precision, our framework is compatible with both deterministic and LLM-based parsers (Zhong et al., 2024; Xing et al., 2025) for environments with noisy natural language outputs. In the scene graph, a node $u \in V_t$ represents an entity (i.e., robot1) and an edge $e \in E_t$ represents a relationship between entities (i.e., cheese1 on-top-of table1). We initialize node features $\mathbf{h}_u^{(0)}$ using a lightweight sentence encoder to capture semantic properties (i.e., entity type, status). To produce a structure-aware state representation of the environment, we process $SG_t$ with a Graph Attention Network (GAT) (Veličković et al., 2018). In embodied planning tasks, spatial relationships are important. For instance, the vertical topology of a burger stack (i.e., patty on-top-of bun) defines dependencies for future actions. The GAT layers preserve local topology by assigning higher attention weights to more relevant neighboring nodes. The node feature $\mathbf{h}_u$ is updated iteratively as in Equation 1. The attention mechanism $\alpha_{u,v}$ enables the encoder to learn these structural attributes.

$$\mathbf{h}_u^{(l)} = \sigma\Big( \sum_{v \in \mathcal{N}(u)} \alpha_{u,v} \mathbf{W}^{(l)} \mathbf{h}_v^{(l-1)} \Big) \quad (1)$$

To derive a fixed-size representation independent of graph size, we apply mean pooling across all node features. We further apply a normalization layer to produce the final graph embedding for $SG_t$: $\mathbf{z}_t = \texttt{BatchNorm}(\texttt{MeanPool}(\{\mathbf{h}_u \mid u \in V_t\}))$. With the help of GAT, the embedding $\mathbf{z}_t$ captures critical structural relationships rather than just statistics of the global scene.

### 3.1.2. STATE-TRANSITION GRAPH (OUTER GRAPH)

While scene graph $SG_t$ captures the agent's instantaneous state at step $t$, the agent's dynamic exploration process and long-term memory are modeled by a higher-level state-transition graph (outer graph $OG$). The $OG$ serves to consolidate and preserve the agent's full exploration trajectory.

Each node $s_t \in OG$ corresponds to a unique abstracted state in the exploration trajectory at step $t$. The node feature of $s_t$ is the GNN-encoded embedding $\mathbf{z}_t$ of scene graph $SG_t$. The edge $e_{(t,t+1)} \in OG$ connecting $s_t$ to $s_{t+1}$ represents the transition induced by the agent's action $a_t$. For a trajectory without loops, the $OG$ forms a state-transition graph (chain):

$$\mathbf{z}_1 \xrightarrow{a_1} \mathbf{z}_2 \xrightarrow{a_2} \cdots \xrightarrow{a_{n-1}} \mathbf{z}_n$$

where $\mathbf{z}_i$ is the embedding of $SG_i$. This sequence of structural state embeddings serves as the agent's core episodic memory, providing essential context for experience retrieval (Sec 3.3) and loop detection (LD, revisiting a structurally identical state). This structural memory is crucial for complex planning because it mitigates context drift (Wu et al., 2025) by aggregating state information into concise, retrievable graph nodes. The corresponding state-transition graphs can then guide future exploration in similar environments.

### 3.1.3. GNN ENCODER OPTIMIZATION

A GNN encoder is trained to minimize a composite loss function $\mathcal{L}$, which is formulated as a weighted sum of a triplet loss term and a uniformity loss term as shown in Equation 2. During training, we sample a batch of triplet embeddings from $OGs$ as anchor-positive-negative ($\mathbf{z}_a, \mathbf{z}_p, \mathbf{z}_n$). The anchor embedding $\mathbf{z}_a$ and the positive embedding $\mathbf{z}_p$ are sampled from the same OG with one step apart, i.e., embeddings $\mathbf{z}_t$ of $SG_t$ and $\mathbf{z}_{t+1}$ of $SG_{t+1}$. We rely on the assumption of environmental coherence: since physical states evolve gradually, temporally adjacent representations should be more proximal to each other than to randomly sampled states from disjoint trajectories. A negative embedding $\mathbf{z}_n$ is randomly sampled from other OGs within the same batch. The parameter $\gamma$ refers to the triplet margin that separates positive and negative pairs. The term $\mathcal{L}_{\text{uniformity}}$ (Wang & Isola, 2020) acts as a regularization term to prevent representational collapse. We define it as the mean of the squared cosine similarity among all unique sample pairs ($\mathbf{z}_i, \mathbf{z}_j$) in Equation 2, which operates on the entire set of $N$ embeddings within batch $Z = \{\mathbf{z}_{a,1}, \ldots, \mathbf{z}_{a,k}, \mathbf{z}_{p,1}, \ldots, \mathbf{z}_{p,k}\}$. Here $N = |Z|$ is twice the total batch size.

$$\begin{aligned}
\mathcal{L} &= \mathcal{L}_{\text{triplet}} + \lambda \mathcal{L}_{\text{uniformity}} \\
&= \mathbb{E}\left[\max\left(0, \|\mathbf{z}_a - \mathbf{z}_p\|_2^2 - \|\mathbf{z}_a - \mathbf{z}_n\|_2^2 + \gamma\right)\right] \\
&\quad + \lambda \left(\frac{1}{N^2} \sum_{i=1}^{N-1} \sum_{j=i+1}^{N} \left(\frac{\mathbf{z}_i \cdot \mathbf{z}_j}{\|\mathbf{z}_i\|\|\mathbf{z}_j\|}\right)^2\right).
\end{aligned} \quad (2)$$

### 3.2. Informed Plan Generation via Bounded Lookahead

Prior work such as ReAct and CoT relies on the LLM to implicitly reason about future states based on the current observation to choose the next action. This mental process often leads to suboptimal, invalid, or unrecoverable actions. To mitigate this in environments where transition dynamics are known, we introduce an optional Bounded Lookahead (BL) module that grounds action generation in explicit 1-step state projections rather than predicted outcomes. The BL module leverages a transition function $\mathcal{T}$ to perform 1-step state projections over the valid action space $A(s)$. In the Robotouille environment, $\mathcal{T}$ is derived from the environment's transition logic (described in PDDL (Garrett et al., 2020)), though our framework is compatible with any learned world model (Li et al., 2024). We denote the set of grounded 1-step projections at step $t$ as $\mathcal{P}(s_t)$:

$$\mathcal{P}(s_t) = \{(a, s')|a \in A(s_t), s' = \mathcal{T}(s_t, a)\}. \quad (3)$$

We define the branching factor of this lookahead as $\epsilon = |A(s_t)|$. This projection operation is computationally feasible because, in task planning environments, the potential action space $A$ at each step is typically constrained to a small, finite action set (Tang & Agrawal, 2020; Luo et al., 2023). The BL module serves as a dynamics verifier rather than a search engine; it provides the immediate post-conditions of actions without computing the subsequent action space $A(s')$. The output $\mathcal{P}(s_t)$ is then injected into the LLM's context alongside the current scene graph $SG_t$, retrieved experience $R_{\mathbf{z}_t}$ based on current $SG_t$ embedding $\mathbf{z}_t$, and the goal $\mathcal{G}$. The selection of the next action $a_{t+1}$ by an LLM is therefore conditioned on this explicit, grounded transition information:

$$a_{t+1} \sim \text{LLM}(\text{Prompt} \mid SG_t, \mathcal{P}(s_t), R_{\mathbf{z}_t}, \mathcal{G}). \quad (4)$$

This transforms the agent from imaginative prediction to discriminative selection. Instead of reasoning over guessed outcomes, the LLM reasons and selects the next action $a_{t+1}$ conditioned on explicit, observable outcomes. For environments where $\mathcal{T}$ is unavailable or the state is partially observable (e.g., in the ALFWorld environment where transition outcomes are not known a priori), $\mathcal{P}(s_t)$ becomes an empty set, and our framework defaults to reasoning using the aggregated graph and past experiences without the BL module. For graph aggregation, we maintain a running update of the scene graph. As new observations are received, the system dynamically adds newly discovered entities and links them to the existing topology. This strategy contrasts sharply with prior works (e.g., ReAct (Yao et al., 2023b), ReCAP (Zhang et al., 2025b)) that rely on extended history windows (64+ conversation exchanges). By using a highly structured, memory-augmented state representation, we reduce overhead, keep the context window tight, and significantly improve both planning efficiency and accuracy.

### 3.3. GiG Memory Retrieval

We leverage LLMs' in-context learning capabilities to use past experiences as in-context examples (Monea et al., 2025)

**Algorithm 1** Conditioned Action Generation with Retrieved Experience as Guidance

---

**Require:** $s_t$: Current observation; $A(s_t)$: Valid actions; $\mathcal{M}$: Memory bank; $\tau$: Retrieval threshold; $GNN$: GNN model; $\mathbf{z}_{t-1}, a_{t-1}$: Previous state node embedding and action; $G_{session}$: Current session graph;

1:  $\mathbf{z}_t \leftarrow GNN(s_t)$           ▷ Embed observation
2:  Prompt $\leftarrow$ ConstructBase$(s_t, A(s_t))$   ▷ Base prompt
3:  **if** $\mathbf{z}_t \in G_{session}$ **then**
4:      $L \leftarrow$ GetLoopPath$(G_{session}, \mathbf{z}_t)$
5:      Append ("Loop Warning", $L$) to Prompt
6:  **end if**
7:  $\mathcal{P}(s_t) \leftarrow \{(a, \mathcal{T}(s_t, a)) \mid \forall a \in A(s_t)\}$
8:  Append ("Lookahead: ", $\mathcal{P}(s_t)$) to Prompt
9:  $(\mathbf{z}_k, d) \leftarrow$ FindClosest$(\mathcal{M}, \mathbf{z}_t)$    ▷ Similarity Search
10: **if** $d < \tau$ **then**
11:     $(\mathcal{R}_{\mathbf{z}_t}, \mathcal{G}_k) \leftarrow$ GetGoalPath$(\mathcal{M}, \mathbf{z}_k)$
12:     Append ("Past Experience", $\mathcal{R}_{\mathbf{z}_t}, \mathcal{G}_k$) to Prompt
13: **end if**
14: $a_t \leftarrow LLM(\text{Prompt})$
15: Update $G_{session}$ with node $\mathbf{z}_t$ and edge $(\mathbf{z}_{t-1}, \mathbf{z}_t, a_{t-1})$
16: **if** Task Success **then**
17:     $\mathcal{M} \leftarrow \mathcal{M} \cup \{G_{session}\}$
18: **end if**
19: **return** $a_t, \mathbf{z}_t$

---

for selecting future actions. Unlike prior work (Kim et al., 2024) that retrieves static subsets of examples, we employ an iterative retrieval strategy. The memory bank $\mathcal{M} = \{\mathcal{E}_j\}_{j=1}^N$ stores a collection of successful task trajectories from past experiences. Each trajectory is structured as a state-transition graph $(OG)$ containing a sequence of GNN-encoded state-action pairs associated with a goal $\mathcal{G}_j$:

$$\mathcal{E}_j = \left( \mathcal{G}_j, [(\mathbf{z}_{j,i}, a_{j,i})]_{i=0}^{T_j} \right). \tag{5}$$

Here $\mathbf{z}_{j,i}$ is the scene graph embedding and $a_{j,i}$ is the action taken at step $i$ of trajectory $j$. The scene graph embedding $\mathbf{z}_t$ acts as a query key to the memory bank $\mathcal{M}$ for retrieving relevant experience memory $\mathcal{R}_{\mathbf{z}_t}$ using:

$$\mathcal{R}_{\mathbf{z}_t} = \begin{cases} \mathcal{S}_{k,m} & \text{if } \min_{j,i} \texttt{Dist}(\mathbf{z}_t, \mathbf{z}_{j,i}) < \tau \\ \emptyset & \text{otherwise.} \end{cases} \tag{6}$$

The retrieved experience $\mathcal{S}_{k,m}$ is a sub-trajectory starting at step $m$, where $k$ corresponds to the best matching experience. The indices $(k, m)$ are chosen to minimize the Euclidean distance $\texttt{Dist}(\mathbf{z}_t, \mathbf{z}_{j,i})$. If a sub-trajectory is within a predefined distance threshold $\tau$, we retrieve the context of that matched state with its subsequent state-transition actions.[1] Because GiG retrieves actions via local scene

---

[1]The value of $\tau$ is investigated empirically in Section 4.2.

---

graph structure similarity, it enables in-domain compositional transfer. For example, the agent can retrieve a successful "cutting" sequence from a sandwich-making experience and apply it to a burger-making task, using the LLM as a semantic filter to adopt or reject these candidate actions. The final retrieved experience includes the subsequent action sequence $A_{k\rightarrow} = [a_{k,m}, a_{k,m+1}, \dots]$. In practice, we retrieve only the single most relevant *immediate* transition, ensuring the agent constantly adapts its reference based on the latest state. The retrieved experience is then formatted into the prompt alongside the current observation, providing the agent with a highly relevant, one-shot example of a past successful trajectory to guide future decision-making. Algorithm 1 summarizes the entire workflow of GiG and the update method for memory bank.

## 4. Experiments

We evaluate the performance of GiG on three embodied planning benchmarks: Robotouille Synchronous, Robotouille Asynchronous (Gonzalez-Pumariega et al., 2025), and ALF-World (Shridhar et al., 2021). Robotouille Synchronous and Asynchronous (Gonzalez-Pumariega et al., 2025) are two benchmarks for long-horizon planning. Both require an agent to accomplish high-level goals (e.g., prepare a lettuce sandwich) by managing strict prerequisites (e.g., cutting before assembly) and resource contention (e.g., waiting for an occupied cutting board). The Asynchronous variant extends this with longer horizons and the additional challenge of concurrency (e.g., cutting while waiting for the patty to be cooked). ALFWorld tests generalization in a partially observable text-based environment, requiring agents to interpret natural language instructions and navigate through diverse, unseen layouts to solve multi-step tasks. Appendix A.1 shows details of each benchmark.

### 4.1. Experiments Setup

We evaluate GiG using a range of open-source and proprietary models of varying scales, including Qwen3-235B (Yang et al., 2025a), Qwen3-30B, DeepSeek-R1 (Guo et al., 2025), Gemini-2.5-Flash (Comanici et al., 2025) , and Gemini-2.5-Flash-Lite. All open-source models are hosted locally on an 8×H100 NVLink-connected server using vLLM. We set the temperature to 0 across all evaluations for reproducibility, with a maximum generation length of 4096 tokens (including reasoning tokens) per LLM call. To build the experience memory bank, we collect 50 successful trajectories using Qwen3-235B across all 10 task types from the training environment seed. GNN embeddings are computed from the scene graphs at each step of every collected trajectory and indexed into the memory bank. We use the Faiss (Douze et al., 2024) library to index the embeddings and build a vector database for efficient similarity search to

mitigate retrieval overhead. The retrieval threshold $\tau$ is set to 0.1 based on the intra-sequence distance distribution analysis in Section 4.2, ensuring the retrieval of topologically similar states while rejecting unrelated noise.

For comparison, we include four baselines that are capable of embodied planning: ReCAP (Zhang et al., 2025b), ReAct (Yao et al., 2023b), ExpeL (Zhao et al., 2024), and CoT (Wei et al., 2022). ReCAP introduces recursive planning with backtracking by maintaining a context tree throughout the planning, marking the state-of-the-art performance on the Robotouille dataset. ReAct has demonstrated strong performance across diverse reasoning and planning benchmarks and remains widely adopted. ExpeL is evaluated on ALFWorld only, following its original evaluation setting, where its distilled experience insights serve as a direct comparison to GiG's dynamically updated graph memory under partial observability and randomized item placements. CoT provides a fundamental baseline that evaluates pure reasoning based planning without environmental feedback. Following ReCAP, we adopt the Pass@1 protocol: each task instance is solved via a single, uninterrupted execution until the task is completed or the maximum step limit is reached, without self-consistency or ensembling.

### 4.2. GNN-Based Scene Graph Encoding

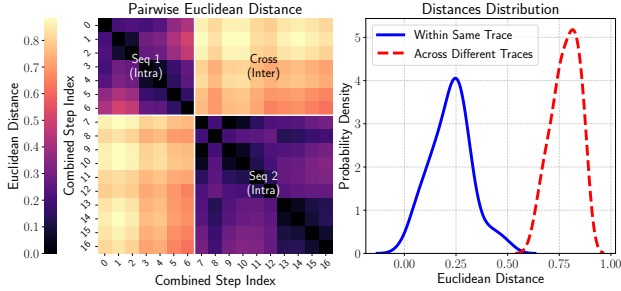

*Figure 3.* Visualization of GNN embedding separation among intra-trace and inter-trace scene graphs.

We use a lightweight GNN to encode each scene graph into a dense representation that captures both object placement and environment topology. To validate the discriminative power of GNN, we study the Euclidean distance between different representations both within a single experience sequence and across distinct sequences. In Figure 3, we illustrate the distribution of embedding distances between two sequences: Seq1 (step 0-6) and Seq2 (step 7-16). The heatmap (left) shows a clear separation between the two sequences: darker colors indicate shorter distances within the same sequence (i.e., Intra), while lighter colors indicate larger distances across sequences (i.e., Inter). This is further supported by the density plot (right), which shows that inter-sequence distances (red dashed line) peak around 0.8, close to the triplet margin $\gamma = 1.0$ used during training. Extended results in

Appendix B further validate this. Note that the off-diagonal dark regions (e.g., between steps 7 and 9) indicate very small distances and correspond to exploration loops, where the agent revisits the same states. Furthermore, adjacent steps within a sequence consistently exhibit distances below 0.1, supporting the choice of threshold $\tau = 0.1$ for retrieving similar experiences. In summary, the clear separation between sequences and tight clustering within sequences show that the GNN-based encoding supports effective scene localization and experience retrieval.

### 4.3. Robotouille Synchronous

We evaluate performance on the Robotouille Synchronous benchmark which spans 10 recipe-completion tasks of increasing horizon length (10-63 steps). For each task, the benchmark provides 10 different starting environments that introduce different contention and object arrangements. To ensure a fair comparison, we align the baselines with GiG by providing them with the same system prompt. Table 1 presents the Pass@1 results across three LLM backbones (detailed results can be found in Appendix C.1). GiG achieves the best performance compared to baselines, surpassing the state-of-the-art ReCAP by up to 22% (Qwen3-235B without memory bank).

*Table 1.* Pass@1 Accuracy on Robotouille Synchronous. Values in parentheses denote the improvement over the best-performing baseline.

| Model | GiG | GiG+Exp | ReCAP | ReAct | CoT |
|---|---|---|---|---|---|
| Qwen3[1] | 93 (+19) | 97 (+23) | 71 | 74 | 7 |
| DeepSeek[2] | 91 (+19) | 88 (+16) | 72 | 53 | 2 |
| Gemini[3] | 92 (+0) | 90 (-2) | 89 | 92 | 34 |

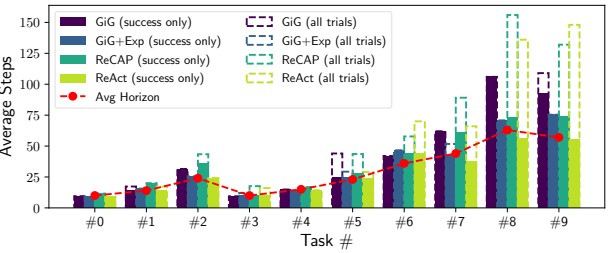

*Figure 4.* Average steps on Robotouille synchronous tasks on Qwen3-235B. Red dots indicate the horizon length of each task type. *success only*: average completion steps of successful attempts. *all trials*: average steps of all attempts.

We further analyze the average steps required to complete each task in Figure 4. For easy to medium difficulty tasks (0-6), all frameworks demonstrate similar average steps,

---

[1]Qwen3-235B-A22B-Instruct-2507

[2]DeepSeek-R1

[3]Gemini-2.5-Flash

closely adhering to the task horizon (red dots). However, for more difficult tasks, we observe that GiG takes slightly more steps in its success trials (solid bars) compared to baselines. The increased number of steps reflects the extra effort to complete tasks on which other baselines fail, suggesting greater robustness on these challenging tasks. Including failed attempts (dashed bars) shifts the results: the baselines require many more steps on average than GiG, highlighting GiG's efficiency across both successes and failures. Augmented with the experience memory (GiG+Exp), the model completes tasks in even fewer steps, demonstrating the effectiveness of our memory architecture. The Pass@1 performance of GiG+Exp remains comparable, likely due to the already strong performance of GiG on the benchmark.

## 4.4. Robotouille Asynchronous

*Table 2.* Pass@1 Accuracy on Robotouille Asynchronous. Values in parentheses denote the improvement over the best-performing baseline.

| Model | GiG | GiG+Exp | ReCAP | ReAct | CoT |
|---|---|---|---|---|---|
| Qwen3[1] | 72 (+37) | 82 (+47) | 35 | 31 | 0 |
| DeepSeek[2] | 59 (+32) | 86 (+59) | 27 | 16 | 0 |
| Gemini[3] | 66 (+6) | 66 (+6) | 21 | 60 | 4 |

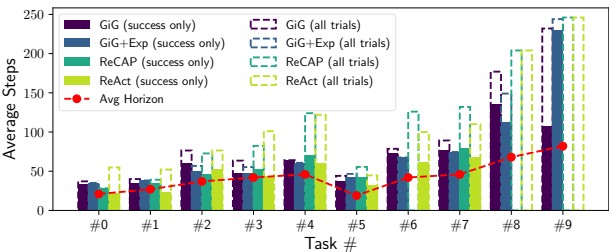

*Figure 5.* Average steps on Robotouille asynchronous tasks on Qwen3-235B. Red dots indicate the average horizon length of each task. *success only*: average completion steps of successful attempts. *all trials*: average steps of all attempts.

The Robotouille Asynchronous benchmark introduces action delays, which allows the agent to interleave other actions while waiting for the background process to finish. Such asynchronous tasks require more complex planning and concurrency management. In Table 2, our evaluation shows that GiG achieves the highest Pass@1 among all frameworks across all LLM models, improving the success rate by up to 37% (Qwen3-235B without memory bank) compared to the best baseline (Details can be found in Appendix C.2). Furthermore, adding experience memory (GiG+Exp) provides an additional 27% improvement on DeepSeek. This highlights that retrieving topologically similar execution traces grounds the agent's decisions in successful history, improving planning performance.

We analyze the step efficiency in Figure 5. In general, GiG

takes slightly more steps with respect to the horizon length while successfully completing more tasks. The augmented experience memory helps reduce the step count in most cases.

## 4.5. ALFWorld with Partial Observation

*Table 3.* Pass@1 Accuracy on ALFWorld. Values in parentheses denote the improvement over the best-performing baseline.

| Model | GiG | ReCAP | ExpeL | ReAct |
|---|---|---|---|---|
| Qwen3[1] | 97 (+6) | 89 | 91 | 61 |
| DeepSeek[2] | 97 (+15) | 82 | 75 | N/A[4] |
| Gemini[3] | 91 (+2) | 86 | 89 | N/A |

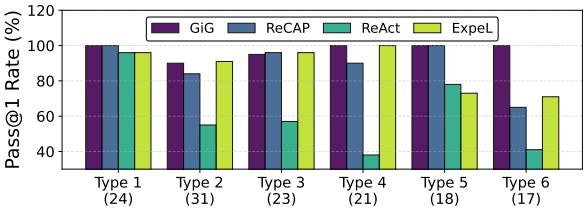

*Figure 6.* Pass rate for each task type on ALFWorld benchmark.

ALFWorld is an embodied planning benchmark for abstract, text-based household tasks. Unlike Robotouille, it features partial observability, requiring the agent to actively explore containers to locate randomized, hidden targets. We evaluate on the standard evaluation set (134 tasks across 6 types). CoT is omitted due to its inability to handle procedurally generated environments without environmental feedback. Furthermore, we exclude experience memory (GiG+Exp) in this setting, as randomized item placements severely limit the transferability of past trajectories. This vulnerability is evident in the ExpeL baseline (Zhao et al., 2024), which relies on retrieving historical text-based insights but struggles to adapt to novel spatial layouts. In contrast, GiG dynamically aggregates exploration observations into a cohesive graph structure. As shown in Table 3, this approach allows GiG to achieve near-perfect performance (97%) on Qwen3 and DeepSeek, outperforming the state-of-the-art methods (detailed by task type in Figure 6 and Appendix C.3).

## 4.6. Experience Memory Plug-in for Small LLMs

We investigate whether experience collected by larger models can benefit smaller models at inference time without fine-tuning. As shown in Table 4, smaller models significantly underperform their larger counterparts across all frameworks, consistent with prior observations (Zhang et al., 2025b). Introducing the experience memory bank recovers performance substantially: GiG(+Exp) achieves absolute gains of +15% with Qwen3-30B and +7% with Gemini-2.5-Flash-Lite, demonstrating that the memory bank serves as a

---

[4]N/A: ReAct failed tasks due to syntax errors or action loops.

model-agnostic plug-in requiring no architectural changes or fine-tuning. Moreover, Figure 7 (excluding Tasks 8 and 9, as no framework successfully completed these with smaller LLMs) shows that using the experience memory not only improves Pass@1 performance but also reduces the steps required for task completion when both success and failed trials are included. Similar to previous observations (Section 4.3 and 4.4), GiG shows a higher step count in success trials because it spends more steps on those longer and more complex tasks where other baselines fail. Detailed task and step analysis can be found in Appendix C.4.

*Table 4.* Pass@1 Accuracy on Robotouille with small models. Values in parentheses denote the improvement over the best-performing baseline.

| Model | GiG | GiG+Exp | ReCAP | ReAct |
|---|---|---|---|---|
| Qwen3[5] | 27 | 42 (+15) | 19 | 28 |
| Gemini[6] | 19 | 26 (+7) | 20 | 20 |

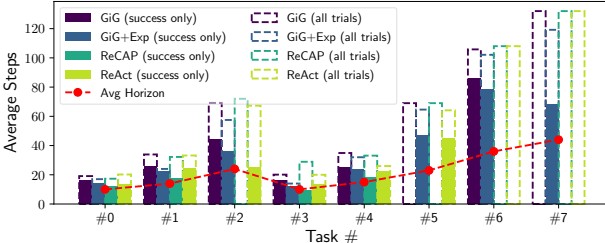

*Figure 7.* Average steps on Robotouille synchronous tasks on Qwen3-30B-A3B. Red dots indicate the average horizon length of each task. *success only*: average completion steps of successful attempts. *all trials*: average steps of all attempts.

## 4.7. Cost Analysis and Scalability

While prior work (Zhang et al., 2025b) conducted cost analysis on dollar cost, the volatility of pricing models, driven by fluctuating rates for input, output, and cached tokens, makes these metrics inconsistent over time. To ensure a standardized comparison, we instead evaluate cost in terms of computation indicators such as FLOPs (Ikram et al., 2025) (Appendix C.6). Unlike baselines such as ReAct and Re-CAP, which process cumulative interaction histories or extensive sliding windows, GiG uses only the immediate state context for each interaction. This architectural distinction significantly reduces computation overhead. As illustrated in Figure 8(b), GiG incurs orders of magnitude fewer FLOPs compared to baselines as the task horizon extends.

We further analyze the scalability of GiG as the number of nodes increases. Figure 8(a) decomposes the cost into graph building latency (sentence encoding latency for the nodes

of the inner graph and edge formation latency of the inner graph) and GNN encoding latency. The GNN encoding latency remains constant regardless of graph size. While the graph construction latency exhibits near-linear growth, it remains consistently in the sub-second regime (<150 ms). Given that LLM decoding spans seconds to tens of seconds with thousands of tokens generated per interaction (as shown in Figure 9), this overhead is negligible in practice.

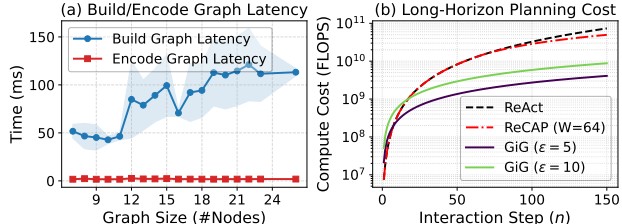

*Figure 8.* (a) GiG graph construction latency remains negligible relative to LLM decoding time. (b) GiG requires orders of magnitude less computation than baselines in long-horizon tasks. $\epsilon$ is the branching factor of BL.

## 4.8. Robustness to Perception Noise

*Table 5.* Robustness under Perception Noise.

| Pass@1 | no-noise | noise/10 steps | noise/5 steps |
|---|---|---|---|
| GiG | 93 | 94 | 93 |
| ReAct | 74 | 67 (-7) | 62 (-12) |
| ReCAP | 71 | 66 (-5) | 62 (-9) |

Real-world planning systems frequently encounter perceptual noise, such as missed objects or false detections. To evaluate our framework's robustness, we introduce controllable noise by randomly adding or removing items from the raw text observations at varying frequencies (e.g., every 5 or 10 steps). As demonstrated in Table 5, GiG exhibits robustness by maintaining a stable Pass@1 rate of 93–94% across all noise frequencies. In contrast, ReAct and ReCAP suffer significant performance degradation, dropping by up to 12% and 9%, respectively. Such resilience stems from the graph architecture, i.e., GiG maintains a persistent, dynamically updated scene graph that naturally filters out transient observation noise during subsequent updates. On the other hand, baseline models treat observations as a flat, concatenated history, which causes noisy observations to permanently remain in the context window along with the correct observations, leading to compounding reasoning errors and hallucinated actions.

## 4.9. Ablation Study

### 4.9.1. EFFECTIVENESS OF EACH MODULE

We conduct an ablation study to evaluate the contribution of each component. As shown in Table 6, experience retrieval

---

[5]Qwen3-30B-A3B-Instruct-2507-FP8
[6]Gemini-2.5-Flash-Lite

alone achieves the strongest single-component performance (95%), as grounding decisions in past successful traces is the primary driver of planning quality. However, high-quality experience collection itself depends on BL, LD and dynamically aggregated scene graph to navigate long-horizon tasks reliably during the collection phase; without them, the memory bank would contain fewer successful trajectories, undermining retrieval effectiveness. The full combination BL+LD+Exp achieves the best performance at 97%. The detailed step analysis is available in Appendix C.5.

*Table 6.* Contribution of each component to Pass@1 performance.

| Metric | BL | LD | Exp | BL+LD | BL+LD+Exp |
|--------|----|----|-----|-------|-----------|
| Pass@1 | 80 | 90 | 95  | 93    | 97        |

### 4.9.2. TOKEN USAGE ACROSS DIFFERENT MODELS

*Table 7.* Comparison of average context window consumption/step (mean ± std) across methods.

| | Avg. Context Consumption per Step (Prefill) | | |
|--------|-------|-------|-------|
| | **GiG** | **ReCAP** | **ReAct** |
| **Tokens** | 12297 (±1840) | 40720 (±26655) | 57357 (±46860) |

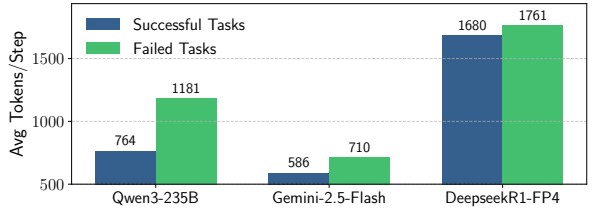

*Figure 9.* Average tokens (reasoning + output) generated per step. Models put more effort for failed tasks.

Table 7 and Figure 9 detail the token efficiency of our framework across both context consumption and generation. In terms of prompt context (prefill), GiG is highly efficient, consuming roughly 30% of the tokens required by ReCAP and 20% of ReAct. This reduction is achieved by iteratively retrieving execution trace based on structural representation similarity along with the aggregated graph rather than appending a growing interaction history. Furthermore, the average tokens generated per step across different backbone models stratified by successful and failed tasks show substantial variance in token usage across the underlying models. Notably, failed tasks consistently lead to longer reasoning traces. This aligns with the intuition that agents intensify their reasoning effort when attempting difficult tasks. This finding also reinforces our earlier observation that higher task completion rates were accompanied by increased average exploration steps.

## 5. Limitation

While GiG demonstrates strong performance in symbolic planning and maintains robustness under perceptual noise, several limitations remain. First, while our structural retrieval mechanism enables in-domain compositional transfer, zero-shot, out-of-domain transfer to entirely unseen task distributions remains an open challenge. Second, our reliance on large language model backbones introduces inherent inference latency. A standard reasoning trace that incurs around 1000 reasoning and output tokens takes several seconds to generate even under a dedicated high-performance serving engine. In highly dynamic physical environments where state changes occur rapidly, this generation latency creates a bottleneck for real-time responsiveness. Third, while prior work (Liu et al., 2025; Ichter et al., 2023) focuses on low-level continuous control and physical execution of embodied agents to resolve actuation and control failure, we explicitly scope GiG to the abstract, symbolic planning layer, leaving the integration of the framework in a real-world robotic system as future work. Finally, deploying LLM-based planners in the real world poses safety-critical risks. Although GiG mitigates hallucinated transitions within the semantic planning layer, large language models can still generate unpredictable edge-case actions. Deploying this framework on real-world robots without formal verification, strict state-space guardrails, and real-time safety monitors could lead to unsafe physical interactions.

## 6. Conclusion

We present GiG, an adaptive task planning framework powered by a GNN-based Graph-in-Graph memory architecture. By grounding current exploration in historical structural experience and bounded look-ahead reasoning, GiG enables proactive decision-making that mitigates context drift in long-horizon tasks. The observed performance improvements highlight the necessity of structured memory representations for robust embodied reasoning in long-horizon planning tasks. Moreover, it is demonstrated that structured experience retrieval can serve as a model-agnostic plug-in and using it as a prior helps improve the performance of less capable LLMs. Future work will expand cross-domain generalization and integrate verifiable safety constraints for physical execution.

## Impact Statement

This paper presents work aimed at advancing embodied AI agents by improving their planning and memory retrieval capabilities in complex task settings. It focuses on structuring and refining past experiences and repurposes them for similar tasks in similar settings in the future. While our proposed graph-in-graph memory structure demonstrates significant gains in complex, long-horizon tasks, we identify inference latency as a current bottleneck for real-time application. We emphasize that future development in this domain should focus on reducing inference latency to enable safe and effective deployment in time-sensitive real-world scenarios, such as autonomous navigation and robotics.

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

# A. Appendix

## A.1. Benchmark Description

### A.1.1. ROBOTOUILLE SYNCHRONOUS MODE

The synchronous mode comprises 10 distinct task types. For each type, 10 starting environments are generated using standard benchmark seeds, yielding a total of 100 evaluation tasks. The detailed goal for each task type is shown in Table 8, with task horizons ranging from 10 to 63 steps.

*Table 8.* Recipe description for robotouille synchronous tasks and their corresponding horizon length. cut: item needs to be cut before being used as an ingredient. Task 7, 8 and 9 require making 2 dishes. The tasks cover a wide range of horizon lengths from 10 to 63.

| | Horizon | Goal |
|---|---|---|
| Task #0 | 10 | table→bread→cheese→bread |
| Task #1 | 14 | table→bread→lettuce(cut)→bread |
| Task #2 | 24 | table→bread→lettuce(cut)→tomato(cut)→bread |
| Task #3 | 10 | table→bottombun→patty→topbun |
| Task #4 | 15 | table→bottombun→patty→cheese→topbun |
| Task #5 | 23 | table→bottombun→patty→cheese→patty→cheese→topbun |
| Task #6 | 36 | table→bottombun→patty→cheese→lettuce(cut)→tomato(cut)→topbun |
| Task #7 | 44 | 2 * table→bread→chicken→lettuce(cut)→bread |
| Task #8 | 63 | 2 * table→bottombun→patty→lettuce(cut)→tomato(cut)→topbun |
| Task #9 | 57 | table→bottombun→patty→cheese→onion(cut)→topbun
table→bread→chicken→lettuce(cut)→tomato(cut)→bread |

### A.1.2. ROBOTOUILLE ASYNCHRONOUS MODE

Similar to the synchronous mode, the asynchronous mode consists of 10 task types. For each type, 10 starting environments are generated using the benchmark-provided seeds, yielding a total of 100 tasks. A key distinction in this mode is the capability to interleave actions; for example, because boiling water requires 3 time steps, the agent can execute other operations during this interval. A detailed description of each task type is provided in Table 9, with task horizons ranging from 19-82 steps.

*Table 9.* Recipe description for robotouille asynchronous tasks and their corresponding horizon length. cut: item needs to be cut before being used as an ingredient. cook: item needs to be cooked before being used as an ingredient. fry: item requires frying before use. boil: item (or mixture of items) requires boiling before use. [...]: ingredients combined in a container before boiling.

| | Horizon | Description |
|---|---|---|
| Task #0 | 21 | table→bread→cheese→chicken(cook)→bread |
| Task #1 | 27 | table→bread→lettuce(cut)→chicken(cook)→bread |
| Task #2 | 37 | table→bread→lettuce(cut)→tomato(cut)→chicken(fry)→bread |
| Task #3 | 42 | table→bottombun→patty→tomato(cut)→topbun, table→potato(cut, fry) |
| Task #4 | 46 | table→bottombun→patty→onion(cut)→cheese→topbun, table→onion(cut, fry) |
| Task #5 | 19 | table→bowl→[water, potato](boiled) |
| Task #6 | 42 | table→bowl→[water, 3 * onion(cut)](boiled) |
| Task #7 | 46 | table→bowl→[water, tomato](boiled)
table→bread→lettuce(cut)→chicken(cook)→bread |
| Task #8 | 68 | table→bowl→[water, tomato(cut), onion(cut)](boiled)
2 * table→bread→chicken(cook)→bread |
| Task #9 | 82 | table→bowl→[water, onion, potato](boiled)
table→bottombun→patty→lettuce(cut)→topbun
table→bread→chicken(cook)→bread
table→onion(cut, fry) |

### A.1.3. ALFWORLD

The evaluation set consists of 134 tasks across 6 types with horizons of 5–15 steps, detailed in Table 10.

*Table 10.* Task type descriptions for ALFWorld tasks in the test set. {obj}: desklight, apple, etc. {recep}: cabinet, cupboard, etc.

| Type | Descriptions |
|------|--------------|
| 1 | pick {obj} and place at {recep} |
| 2 | pick&clean {obj} and place at {recep} |
| 3 | pick&heat {obj} and place at {recep} |
| 4 | pick&cool {obj} and place at {recep} |
| 5 | look/examine {obj} under {obj} |
| 6 | pick two {obj} and place at {recep} |

## B. GNN Model

### B.1. Embedding Separation Analysis & choice of $\tau$

We further analyze the efficacy of our trained GNN in discriminating between scene embeddings across distinct tasks while maintaining high cohesion within individual trajectories. Figure 10 (left) visualizes the pairwise Euclidean distances between embeddings from two disparate sequences (Task 0 and Task 3) in the Robotouille Synchronous environment. The resulting heatmap reveals a distinct block-diagonal structure, confirming that intra-sequence embeddings remain tightly clustered (low distance) while inter-sequence embeddings are clearly separated (high distance). However, Figure 10 (right) illustrates a pair of sequences from the Asynchronous environment, where this boundary is less defined. As observed, when the environment's horizon length increases, the clear separation between distinct trajectories begins to blur. This phenomenon underscores the importance of our distance threshold, $\tau$, which must be explicitly calibrated to filter out irrelevant context and isolate structurally similar sub-graphs when sequence boundaries degrade. We empirically choose $\tau = 0.1$ based on our results, as this value clearly separates consecutive states while filtering unrelated trajectories.

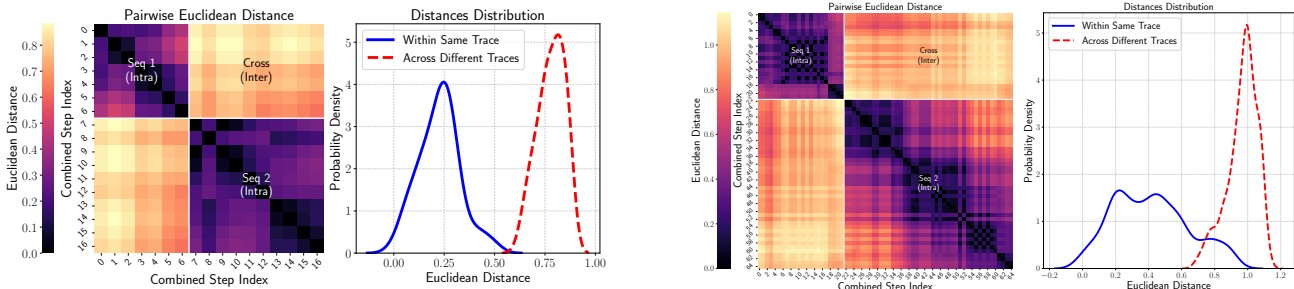

*Figure 10.* GNN embedding separation analysis for Robotouille Synchronous: intra- vs. inter-trace (left) and Robotouille Asynchronous: intra- vs. inter-trace (right) separation.

### B.2. GNN Architecture and Training Config

The encoder consists of two GATConv layers. The first layer maps input features to a hidden dimension using four attention heads, followed by BatchNorm. The second layer projects the concatenated head outputs back using a single attention head, followed by BatchNorm. A final linear layer projects the embeddings to an output dimension of 64. To mitigate overfitting, we apply a dropout rate of 0.6 across all layers. The network is trained for 200 epochs using a dataset of 50 exploration sequences collected from the Robotouille environment using the training seed. The uniformity loss weight $\lambda$ is set to 0.5.

# C. Detailed Results

Unless specified, by default, GiG has bounded lookahead and loop detection enabled.

## C.1. Results on Robotouille Synchronous

We report detailed per-task results for all three LLM backbones on Robotouille Synchronous. Step analysis is provided for Qwen3-235B; Gemini-2.5-Flash and DeepSeek-R1 follow similar trends and step results are omitted for brevity. CoT steps are not reported as the method generates a single plan without environment interaction, resulting in a fixed step count of 1.

*Table 11.* Task completion rate and step comparison for Robotouille **synchronous** tasks with **Qwen3-235B-A22B-Instruct-2507-FP8**. Each task is repeated ten times with different random seeds. A multiplier of three is applied to cap the maximum number of steps, following prior literature (Zhang et al., 2025b).

| Task # (Steps) | GiG | | GiG (+Exp) | | ReCAP | | ReAct | | CoT | |
|---|---|---|---|---|---|---|---|---|---|---|
| | Pass (%) | Steps | Pass (%) | Steps | Pass (%) | Steps | Pass (%) | Steps | Pass (%) | Steps |
| Task #0 (10) | 100 | $9.5 \pm 3.8$ | 100 | $8.9 \pm 2.3$ | 100 | $11.6 \pm 3.3$ | 100 | $8.9 \pm 2.1$ | 30 | - |
| Task #1 (14) | 90 | $17.3 \pm 8.5$ | 100 | $15.6 \pm 4.2$ | 100 | $20.0 \pm 6.9$ | 90 | $17.2 \pm 8.8$ | 20 | - |
| Task #2 (24) | 100 | $31.5 \pm 7.2$ | 100 | $25.5 \pm 4.1$ | 80 | $43.4 \pm 17.6$ | 100 | $24.1 \pm 3.2$ | 0 | - |
| Task #3 (10) | 100 | $9.4 \pm 2.2$ | 90 | $12.1 \pm 6.7$ | 60 | $17.6 \pm 10.2$ | 70 | $16 \pm 9.4$ | 0 | - |
| Task #4 (15) | 100 | $14.9 \pm 4.4$ | 100 | $14 \pm 2.8$ | 100 | $16.8 \pm 2.8$ | 100 | $14.3 \pm 2.2$ | 10 | - |
| Task #5 (23) | 60 | $44 \pm 25.4$ | 90 | $29.3 \pm 13.8$ | 60 | $43.6 \pm 21.3$ | 90 | $29 \pm 14$ | 10 | - |
| Task #6 (36) | 100 | $42 \pm 8.83$ | 100 | $46.6 \pm 9.1$ | 80 | $57.8 \pm 27.1$ | 60 | $70 \pm 30.6$ | 0 | - |
| Task #7 (44) | 100 | $62 \pm 39.2$ | 90 | $51.7 \pm 21.4$ | 60 | $89.1 \pm 35.0$ | 70 | $66 \pm 39.6$ | 0 | - |
| Task #8 (63) | 100 | $106 \pm 34.4$ | 100 | $70.8 \pm 11.2$ | 30 | $156 \pm 53.4$ | 40 | $136 \pm 65.6$ | 0 | - |
| Task #9 (57) | 80 | $109 \pm 37.4$ | 100 | $75.4 \pm 12.9$ | 40 | $132 \pm 47.8$ | 20 | $148 \pm 46.7$ | 0 | - |
| Average | 93 | - | 97 | - | 71 | - | 74 | - | 7 | - |

*Table 12.* Task completion rate for Robotouille **synchronous** tasks with **Gemini-2.5-Flash**. Each task is repeated ten times with different random seeds. A multiplier of three is applied to cap the maximum number of steps, following prior literature (Zhang et al., 2025b).

| | Task | #0 | #1 | #2 | #3 | #4 | #5 | #6 | #7 | #8 | #9 | Avg |
|---|---|---|---|---|---|---|---|---|---|---|---|---|
| GiG(+Exp) | Pass(%) | 100 | 100 | 100 | 100 | 100 | 90 | 100 | 80 | 70 | 60 | 90 |
| GiG | Pass(%) | 100 | 100 | 100 | 100 | 100 | 100 | 100 | 70 | 60 | 90 | 92 |
| ReCAP | Pass(%) | 100 | 100 | 100 | 60 | 100 | 100 | 90 | 80 | 70 | 90 | 89 |
| ReAct | Pass(%) | 100 | 100 | 100 | 100 | 100 | 90 | 80 | 100 | 60 | 90 | 92 |
| CoT | Pass(%) | 60 | 40 | 30 | 30 | 30 | 60 | 50 | 10 | 20 | 10 | 34 |

*Table 13.* Task completion rate for Robotouille **synchronous** tasks with **DeepSeek-R1-FP4**. Each task is repeated ten times with different random seeds. A multiplier of three is applied to cap the maximum number of steps, following prior literature (Zhang et al., 2025b).

| | Task | #0 | #1 | #2 | #3 | #4 | #5 | #6 | #7 | #8 | #9 | Avg |
|---|---|---|---|---|---|---|---|---|---|---|---|---|
| GiG(+Exp) | Pass(%) | 100 | 100 | 90 | 90 | 100 | 90 | 80 | 100 | 80 | 50 | 88 |
| GiG | Pass(%) | 100 | 100 | 100 | 60 | 100 | 90 | 100 | 100 | 100 | 60 | 91 |
| ReCAP | Pass(%) | 90 | 80 | 90 | 50 | 100 | 80 | 80 | 60 | 50 | 40 | 72 |
| ReAct | Pass(%) | 90 | 70 | 60 | 70 | 80 | 20 | 60 | 40 | 20 | 20 | 53 |
| CoT | Pass(%) | 0 | 10 | 0 | 10 | 0 | 0 | 0 | 0 | 0 | 0 | 2 |

## C.2. Results on Robotouille Asynchronous

We report detailed per-task results for all three LLM backbones on Robotouille Asynchronous. Step analysis is provided for Qwen3-235B; Gemini-2.5-Flash and DeepSeek-R1 follow similar trends and step results are omitted for brevity. CoT steps are not reported as the method generates a single plan without environment interaction, resulting in a fixed step count of 1.

*Table 14.* Task completion rate and step comparison for Robotouille **asynchronous** tasks with **Qwen3-235B-A22B-Instruct-2507-FP8**. Each task is repeated ten times. A multiplier of three caps the maximum steps (Zhang et al., 2025b).

| Task # | GiG | | GiG(+Exp) | | ReCAP | | ReAct | | CoT | |
|---|---|---|---|---|---|---|---|---|---|---|
| (Steps) | Pass (%) | Steps | Pass (%) | Steps | Pass (%) | Steps | Pass (%) | Steps | Pass (%) | Steps |
| Task #0 (21) | 90 | $37.1 \pm 13.6$ | 100 | $35.1 \pm 13.3$ | 100 | $28.2 \pm 6.9$ | 20 | $54.8 \pm 16.6$ | 0 | — |
| Task #1 (27) | 90 | $40 \pm 21.1$ | 100 | $38.2 \pm 15.7$ | 90 | $39.2 \pm 21.6$ | 50 | $52.2 \pm 28.9$ | 0 | — |
| Task #2 (37) | 70 | $76.4 \pm 23.7$ | 90 | $56.8 \pm 21.1$ | 60 | $72.7 \pm 32.2$ | 60 | $76.4 \pm 33.1$ | 0 | — |
| Task #3 (42) | 80 | $63.5 \pm 31.9$ | 90 | $55.3 \pm 24.5$ | 60 | $82.4 \pm 36.2$ | 30 | $101 \pm 34.9$ | 0 | — |
| Task #4 (46) | 100 | $64.1 \pm 9.9$ | 100 | $60.7 \pm 10.4$ | 20 | $124 \pm 27.2$ | 20 | $122 \pm 31.5$ | 0 | — |
| Task #5 (19) | 70 | $44.1 \pm 12.1$ | 70 | $46.6 \pm 8.6$ | 10 | $55.5 \pm 4.5$ | 50 | $44.7 \pm 12.7$ | 0 | — |
| Task #6 (42) | 90 | $78.6 \pm 20.4$ | 100 | $67.6 \pm 7.1$ | 0 | $126 \pm 0$ | 40 | $100 \pm 33.0$ | 0 | — |
| Task #7 (46) | 80 | $89.2 \pm 26.3$ | 100 | $74.5 \pm 22.5$ | 10 | $132 \pm 17.3$ | 40 | $110 \pm 35.9$ | 0 | — |
| Task #8 (68) | 40 | $177 \pm 38.9$ | 60 | $149 \pm 46.6$ | 0 | $204 \pm 0$ | 0 | $204 \pm 0$ | 0 | — |
| Task #9 (82) | 10 | $232 \pm 41.7$ | 10 | $244 \pm 4.8$ | 0 | $246 \pm 0$ | 0 | $246 \pm 0$ | 0 | — |
| **Average** | **72** | — | **82** | — | 35 | — | 31 | — | 0 | — |

*Table 15.* Task completion rate for Robotouille **asynchronous** tasks with **Gemini-2.5-Flash**. Each task is repeated ten times with different random seeds. A multiplier of three is applied to cap the maximum number of steps, following prior literature (Zhang et al., 2025b).

| | Task | #0 | #1 | #2 | #3 | #4 | #5 | #6 | #7 | #8 | #9 | Avg |
|---|---|---|---|---|---|---|---|---|---|---|---|---|
| GiG(+Exp) | Pass(%) | 90 | 70 | 80 | 90 | 90 | 60 | 70 | 90 | 20 | 0 | 66 |
| GiG | Pass(%) | 90 | 100 | 70 | 100 | 70 | 60 | 60 | 90 | 20 | 0 | 66 |
| ReCAP | Pass(%) | 30 | 60 | 80 | 10 | 10 | 10 | 10 | 0 | 0 | 0 | 21 |
| ReAct | Pass(%) | 100 | 100 | 90 | 60 | 70 | 70 | 60 | 40 | 10 | 0 | 60 |
| CoT | Pass(%) | 10 | 30 | 0 | 0 | 0 | 0 | 0 | 0 | 0 | 0 | 4 |

*Table 16.* Task completion rate for Robotouille **asynchronous** tasks with **DeepSeek-R1-FP4**. Each task is repeated ten times with different random seeds. A multiplier of three is applied to cap the maximum number of steps, following prior literature (Zhang et al., 2025b).

| | Task | #0 | #1 | #2 | #3 | #4 | #5 | #6 | #7 | #8 | #9 | Avg |
|---|---|---|---|---|---|---|---|---|---|---|---|---|
| GiG(+Exp) | Pass(%) | 100 | 90 | 100 | 80 | 100 | 100 | 80 | 100 | 80 | 30 | 86 |
| GiG | Pass(%) | 80 | 90 | 80 | 80 | 80 | 90 | 50 | 40 | 0 | 0 | 59 |
| ReCAP | Pass(%) | 60 | 70 | 40 | 40 | 30 | 10 | 20 | 0 | 0 | 0 | 27 |
| ReAct | Pass(%) | 10 | 20 | 30 | 20 | 20 | 30 | 20 | 10 | 0 | 0 | 16 |
| CoT | Pass(%) | 0 | 0 | 0 | 0 | 0 | 0 | 0 | 0 | 0 | 0 | 0 |

## C.3. Results on ALFWorld

We report detailed per-task results across all six task types for all three LLM backbones on ALFWorld. ReAct results are unavailable for Gemini and DeepSeek due to these models producing malformed action strings not recognized by the ALFWorld environment, resulting in repeated invalid action loops. While GiG achieves strong overall performance, per-task results reveal that different models exhibit varying strengths across task types. Tasks requiring precise multi-step state tracking (Type 3: pick&heat, Type 4: pick&cool) show higher variance across models, while simpler pick-and-place tasks (Type 1) remain consistently strong across all backbones.

*Table 17.* Task pass rate (%) on ALFWorld with **Qwen3-235B-A22B-Instruct-2507-FP8**.

| Type (# tasks) | GiG | ReCAP | ExpeL | ReAct |
|---|---|---|---|---|
| Type 1 (24) | 100 | 100 | 100 | 96 |
| Type 2 (31) | 90 | 84 | 84 | 55 |
| Type 3 (23) | 96 | 96 | 96 | 57 |
| Type 4 (21) | 100 | 90 | 100 | 38 |
| Type 5 (18) | 100 | 100 | 83 | 78 |
| Type 6 (17) | 100 | 65 | 82 | 41 |
| Average (%) | **97** | 89 | 91 | 61 |

*Table 18.* Task pass rate (%) on ALFWorld with **Gemini-2.5-Flash**. ReAct frequently stuck at invalid actions.

| Type (# tasks) | GiG | ReCAP | ExpeL | ReAct |
|---|---|---|---|---|
| Type 1 (24) | 92 | 100 | 96 | N/A |
| Type 2 (31) | 90 | 80 | 90 | N/A |
| Type 3 (23) | 100 | 82 | 96 | N/A |
| Type 4 (21) | 71 | 95 | 100 | N/A |
| Type 5 (18) | 100 | 67 | 72 | N/A |
| Type 6 (17) | 100 | 100 | 71 | N/A |
| Average (%) | **91** | 86 | 89 | N/A |

*Table 19.* Task pass rate (%) on ALFWorld with **DeepSeek-R1-FP4**. ReAct frequently stuck at invalid actions.

| Type (# tasks) | GiG | ReCAP | ExpeL | ReAct |
|---|---|---|---|---|
| Type 1 (24) | 100 | 100 | 87 | N/A |
| Type 2 (31) | 87 | 65 | 84 | N/A |
| Type 3 (23) | 100 | 78 | 87 | N/A |
| Type 4 (21) | 100 | 100 | 76 | N/A |
| Type 5 (18) | 100 | 94 | 89 | N/A |
| Type 6 (17) | 100 | 59 | 12 | N/A |
| Average (%) | **97** | 82 | 75 | N/A |

## C.4. Results on Small Models

We report detailed per-task results for smaller models on Robotouille Synchronous. All frameworks suffer significant performance degradation compared to their larger counterparts, with most methods failing entirely on longer-horizon tasks (#5–#9). GiG(+Exp) consistently outperforms all baselines on both Qwen3-30B and Gemini-2.5-Flash-Lite, demonstrating that experience retrieval provides benefit when model capability is limited. The step analysis (Table 22) shows that GiG(+Exp) incurs higher average steps than ReCAP and ReAct on simpler tasks, consistent with our earlier observation that GiG succeeds on harder tasks where baselines fail entirely.

*Table 20.* Task completion rate for Robotouille Synchronous tasks with **Qwen3-30B**.

| | Task | #0 | #1 | #2 | #3 | #4 | #5 | #6 | #7 | #8 | #9 | Avg |
|---|---|---|---|---|---|---|---|---|---|---|---|---|
| GiG(LD+EXP) | Pass(%) | 80 | 90 | 40 | 90 | 60 | 20 | 20 | 20 | 0 | 0 | 42 |
| GiG(LD) | Pass(%) | 80 | 50 | 10 | 70 | 50 | 0 | 10 | 0 | 0 | 0 | 27 |
| ReCAP | Pass(%) | 70 | 40 | 0 | 20 | 40 | 0 | 0 | 0 | 0 | 0 | 17 |
| ReAct | Pass(%) | 60 | 50 | 10 | 60 | 80 | 20 | 0 | 0 | 0 | 0 | 28 |
| CoT | Pass(%) | 0 | 0 | 0 | 30 | 10 | 10 | 0 | 0 | 0 | 0 | 5 |

*Table 21.* Task completion rate for Robotouille Synchronous tasks with **Gemini-2.5-Flash-Lite**.

| | Task | #0 | #1 | #2 | #3 | #4 | #5 | #6 | #7 | #8 | #9 | Avg |
|---|---|---|---|---|---|---|---|---|---|---|---|---|
| GiG(+Exp) | Pass(%) | 70 | 50 | 10 | 50 | 80 | 0 | 0 | 0 | 0 | 0 | 26 |
| GiG | Pass(%) | 80 | 30 | 0 | 40 | 40 | 0 | 0 | 0 | 0 | 0 | 19 |
| ReCAP | Pass(%) | 60 | 50 | 0 | 50 | 40 | 0 | 0 | 0 | 0 | 0 | 20 |
| ReAct | Pass(%) | 80 | 20 | 0 | 60 | 30 | 10 | 0 | 0 | 0 | 0 | 20 |

*Table 22.* Average successful steps on Robotouille tasks (**Qwen3-30B**). '–' indicates 0% success rate.

| Method | #0 | #1 | #2 | #3 | #4 | #5 | #6 | #7 | #8 | #9 |
|---|---|---|---|---|---|---|---|---|---|---|
| GiG | 16.4 | 25.8 | 44.0 | 15.8 | 24.8 | — | 86.0 | — | — | — |
| GiG(+Exp) | 13.8 | 22.4 | 35.8 | 12.2 | 23.3 | 47.0 | 78.5 | 68.0 | — | — |
| ReCAP | 11.8 | 17.5 | — | 9.4 | 18.3 | — | — | — | — | — |
| ReAct | 13.6 | 24.4 | 25.0 | 13.3 | 22.0 | 44.5 | — | — | — | — |

### C.5. Ablation Step analysis

*Table 23.* Task completion rate and step comparison for Robotouille Synchronous tasks with **Qwen3-235B-A22B-Instruct-2507-FP8** with different components enabled.

|  | Task | #0 | #1 | #2 | #3 | #4 | #5 | #6 | #7 | #8 | #9 | Avg |
|---|---|---|---|---|---|---|---|---|---|---|---|---|
| GiG(+LD only) | Pass (%) | 100 | 100 | 100 | 100 | 100 | 20 | 100 | 100 | 90 | 90 | 90 |
|  | Steps | 9.9 | 16.2 | 32.7 | 10.3 | 13.9 | 65.2 | 38.5 | 59.0 | 119.3 | 106.3 | — |
| GiG(+BL only) | Pass (%) | 100 | 80 | 90 | 100 | 100 | 60 | 90 | 90 | 70 | 20 | 80 |
|  | Steps | 9.5 | 22.4 | 38.4 | 9.7 | 15.3 | 44.2 | 48.4 | 62.3 | 123.5 | 148.3 | — |
| GiG(+Exp only) | Pass (%) | 90 | 100 | 100 | 90 | 100 | 90 | 100 | 100 | 100 | 80 | 95 |
|  | Steps | 11.8 | 16.7 | 34.9 | 12.1 | 15.1 | 34.3 | 38.8 | 52.2 | 77.7 | 82.3 | — |
| GiG(+LD+BL) | Pass (%) | 100 | 90 | 100 | 100 | 100 | 60 | 100 | 100 | 100 | 80 | 93 |
|  | Steps | 9.5 | 17.3 | 31.2 | 9.4 | 14.9 | 44.0 | 42.0 | 62.1 | 105.7 | 108.6 | — |
| GiG(+LD+BL+Exp) | Pass (%) | 100 | 100 | 100 | 90 | 100 | 90 | 100 | 90 | 100 | 100 | 97 |
|  | Steps | 8.9 | 15.6 | 25.5 | 12.1 | 14 | 29.3 | 46.6 | 51.7 | 70.8 | 75.4 | — |

### C.6. Compute Cost Analysis with Bounded Lookahead

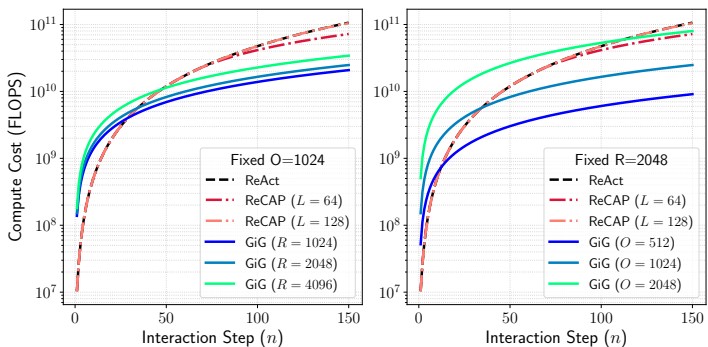

*Figure 11.* Compute cost analysis for GiG and baselines under fixed O (observation tokens)(left) and fixed R (reasoning+output tokens) (right) with a fixed available action space (branching factor) $\epsilon = 10$.

To show that the additional bounded lookahead simulation in the query prompt does not incur significant costs in long horizon settings, we analyze the attention mechanism's cost. Following (Ikram et al., 2025), we define the following notation:

- $S$: System prompt length.

- $O$: Observation prompt length, assumed to be constant (mean across interactions).

- $R$: Reasoning and output token length, assumed to be constant (mean across interactions).

- $\epsilon$: Branching factor, representing the number of valid actions available at each interaction.

We first analyze the attention compute cost for ReAct and ReCAP baselines. The cost is composed of two parts in each interaction: the prefill (system + initial observation prompt) and the decoding (generating thinking and output tokens). We assume a standard KV-cache implementation where prior states are preserved given sufficient GPU memory, as seen in serving engines such as vLLM (Kwon et al., 2023).

The compute cost for the $k$-th interaction is derived as follows:

**1st Interaction ($k = 1$):** The model processes the system prompt and the initial observation, followed by the generation of R tokens. In the prefill phase, the input sequence attends to itself, resulting in a cost proportional to $(S + O)^2$. During decoding, each new token attends to the entire prefix (system prompt + observation) as well as all preceding tokens generated in the current step. The total cost is given by

$$C_1 = \underbrace{(S + O)^2}_{\text{Prefill}} + \underbrace{\sum_{i=1}^{R}(S + O + i)}_{\text{Decoding}} \tag{7}$$

**2nd Interaction ($k = 2$):** The new observation tokens attend to the history $S + O + R$ preserved from the first interaction, followed by the generation of R new output tokens [2]

$$C_2 = \underbrace{(S + O + R) \cdot O}_{\text{Prefill}} + \underbrace{\sum_{i=1}^{R}(S + 2O + R + i)}_{\text{Decoding}} \tag{8}$$

**$k$-th Interaction (General Case):** For the $k$-th step (where $k > 1$), the history length is $S + (k - 1)(O + R)$.

$$C_k = [S + (k - 1)(O + R)] \cdot O + \sum_{i=1}^{R}[S + kO + (k - 1)R + i] \tag{9}$$

**Total Compute Cost:** Summing over $N$ interactions, the total attention cost $C_{total}$ is:

$$C_{total} = C_1 + \sum_{k=2}^{N} C_k \tag{10}$$

Next, we analyze the compute cost for **GiG**. Unlike **ReAct** and **ReCAP**, GiG incorporates $\epsilon$ additional observations derived from the lookahead simulation.

**1st Interaction:** The model processes the system prompt and the initial observation, followed by the generation of R tokens. In the prefill phase, the input sequence attends to itself, resulting in a cost proportional to $(S + \epsilon O)^2$. A branching factor $\epsilon$ is included to account for the additional one-step simulated lookahead results.

$$C_k = (S + \epsilon O)^2 + \sum_{i=1}^{R}(S + \epsilon O + i) \tag{11}$$

**2nd Interaction:** The model processes the new observation alongside the lookahead simulations $\epsilon O$ by attending to $(S + \epsilon O + R_1)$ from the immediate previous interaction. It then generates R tokens. Crucially, we retain only the most recent step to track progress, rather than accumulating the full history. While a concise textual summary could theoretically suffice, our experimental results indicated that certain models (e.g., Gemini-2.5-Flash, DeepSeek-R1-FP4) struggle to adhere to the required information extraction formats. Consequently, we keep the full reasoning and output information from the preceding step rather than extract specific information from it for experiments with DeepSeek and Gemini.

$$C_2 = (S + \epsilon O + R_1) \cdot \epsilon O + \sum_{i=1}^{R}(S + \epsilon O + R_1 + i) \tag{12}$$

---

[2]We omit the self-attention cost for the new observation tokens $O^2$ as it is negligible relative to the accumulated history in long-horizon interactions.

**kth Interaction (General Case):**    The model only attends to the $k-1$th interaction history while processing the new observation along with the one-step lookahead observation $\epsilon O$.

$$C_k = (S + \epsilon O + R_{k-1}) \cdot \epsilon O + \sum_{i=1}^{R}(S + \epsilon O + R_{k-1} + i) \tag{13}$$

**Total Compute Cost:**    Summing over $N$ interactions, the total computation cost is shown below, where $C_k$ remains unchanged with respect to the growing interactions since we only use the most recent interaction throughout the process.

$$C_{total} = C_1 + \sum_{k=2}^{N} C_k \tag{14}$$

Based on this analysis, we compare the attention computation costs of GiG against baseline methods under varying conditions. Figure 11 illustrates that while baselines incur lower computational costs for short-horizon planning ($< 25$ steps), GiG demonstrates significantly superior scalability, requiring fewer resources for long-horizon tasks.

## D. Prompt used in Robotouille

---

**System Prompt: Robotouille Task Planning**

```
system: |
  You are an agent exploring a game environment with a goal to achieve. You will
  propose an action in the current state to make progress toward the goal. Follow the
  rules carefully since the environment may have constraints that do not align with
  the real world.
instructions: |
  You must propose an action given the current observation, progress, past experience
  and valid actions and the last reasoning and action taken in the environment. Make
  use of the environment simulation to guide your next action, this simulation tells
  you the result of taking each valid action.

  You will receive the initial state and the goal as follows:
  Optional[Error Feedback: ...]
  Observation: ...
  Valid Actions: ...
  Goal: ...
  Environment simulation: ...
  Last Step Summary: ...
  Optional[Past Experience: ...]

  where
  - 'Observation' contains state information about objects in the environment and the
  goal
  - 'Valid Actions' is the list of actions you can take in the current state
  - 'Goal' is the request that needs to be fulfilled, such as 'making a hamburger'
  - 'Error Feedback' includes feedback about an invalid action taken in a previous
  interaction
    - This feedback is automated and shows if the action is either syntactically
    incorrect or does not exist in the valid actions list
    - This feedback does not check for semantic correctness and should neither
    reinforce nor discourage the current strategy
    - If the environment indicates that the previous action resulted in an error,
    then any assumed progress from that failed action is incorrect. You must revert
    the Current Progress: back to what it was before the failed action was attempted.
  - 'Last Step Summary' contains a short summary of the reasoning in previous step.
```

– 'Past Experience' includes valuable experience learned from past actions. If past experience exists, pay close attention to the actions taken in the past, especially if it leads to a loop.
– 'Environment simulation' is a list of environment observations resulting from taking each potential valid action from the current environment

Always format your response as follows:
Reasoning: ...
Action: ...
Summary: ...
Current Progress: ...

where:
– 'Reasoning' includes reasons about the action you will propose to take next
  – **Identify Goal:** Read the `Goal:` line. Determine the target recipe (e.g., sandwich, hamburger) and the required ingredients (e.g., lettuce, tomato).
  – **Determine Required Stack Order:** Based on the recipe knowledge and Goal, construct the specific bottom-to-top stack needed for the goal. For a "sandwich with lettuce and tomato," the default order implies the stack: **bread -> ingredients(cut tomato, cut lettuce) -> bread**.
  – **Adopt a Flexible Strategy:** Do not prepare *all* ingredients before starting to stack. It's often better to **process an ingredient, then immediately place it** on the stack (if it's the correct next layer) or move it to a temporary clean station.
  – **Determine ingredient state:** Make sure the ingredients are prepared (i.e., cut, cooked, fried, boiled) according to the goal before placing. Only lettuce, tomato, potato can be cut.
  – **Check Workspace Before Processing:** Before planning to prepare an ingredient (e.g., cut lettuce on board1, cook patty on stove1):
      * Check if the required workspace (board, fryer, stove, sink) is free.
      * If the workspace is **occupied** (e.g., by a cut tomato), you **must first plan to move the occupying item** off the workspace. Move it either to the final sandwich stack (if it's the correct next layer) or to a temporary clean station.
  – **Identify robot state:** If the robot is currently holding something not needed, put it at an empty station that does not interfere with the current recipe.
  – Include a complete step by step action plan toward the goal to justify the next action you'll propose to take
– 'Action' is the action you propose to take. This action must be chosen from the **Valid Actions** provided, with the **exact wording**.
– 'Summary' is a short summary of the environment description of objects in the current environment and the action you propose towards the goal. Keep the summary short; aim for fewer than 150 words.
– 'Current Progress' is the progress of the current recipe after the current action is performed, or the existing arrangement that can be directly used. For example, if the current environment has bread1 on table5, and we want to make a cheese sandwich whose recipe is table->bread->cheese->bread. We can directly use the existing table5->bread1 as the base, and it should be the current progress. If the current action does not lead to any progress, the Current Progress remains the same. Current Progress should not include actions (i.e., cut, cook)

Below is a description of the environment:
You are a robot in a kitchen environment. The objects in the kitchen and your goal are described in the Observation. The various types of objects in the kitchen include
– Station: A location in the kitchen where you can perform special actions, e.g. cooking or cutting or frying or boiling, if an item occupies the station, you can move it somewhere else.
– Item: An object that can be picked up and potentially used in a Station.
– Player: Robots, including you, that are present in the kitchen.
– Container: An object that can hold meals, e.g. a pot or a pan.
– Meal: A mixture of ingredients contained within a Container.

The rules of the environment are as follows:
- A Player can only hold a single item at a time. If you want to get an item but are already holding something, you can move to an empty station (e.g., table, fryer, sink) and place the holding item there.
- An item must be placed on a Station to perform an action on it.
- A Station must contain a single Item to perform an action on it.
- Items can only be stacked on top of one another.
- A Container must contain a Meal to have items added to it.
- A Meal can be transferred between Containers.
- An item may take several steps to cook, once you start the cook action, you can leave it and do other tasks while it is cooking.
- An item may take several steps to fry, once you start the fry action, you can leave it and do other tasks while it is frying.
- An item may take several steps to boil, once you start the boil action, you can leave it and do other tasks while it is boiling.

The goal of this environment is to satisfy a human's request, such as 'make me a hamburger'. These goals are intentionally underspecified, so common sense reasoning is required to complete them. Specifically, it is important to consider
- the minimal ingredients required to satisfy the request
- any preparation steps for the ingredients like cooking, cutting, etc.
- if all tables are occupied and cannot form a base, try to move the ingredients around to form a base first.

When the goal is achieved or a time limit is reached, the environment will end.

Follow this recipe guide to learn how to make food in Robotouille:
Sandwich – A slice of bread, stacked on prepared(cut,cooked) ingredients, stacked on another slice of bread. bread is *NOT* the same as a bun.
  i.e. lettuce sandwich – A slice of bread, stacked on cut lettuce, stacked on another slice of bread.
Hamburger – A top bun, stacked on prepared ingredients (optional), stacked on a cooked patty, stacked on a bottom bun.
  i.e. lettuce burger – A top bun, stacked on cut lettuce, stacked on a cooked patty, stacked on a bottom bun.
Soup – A pot is filled with water, then boiled while ingredients are added, then served in a bowl when ready.
  i.e. tomato soup – A pot filled with water under the sink, then put on the stove, add a whole tomato into the pot, boil, then serve in a bowl.

