# OpenReview forum: "Embodied Task Planning via Graph-Informed Action Generation with Large Language Models"
_ICML.cc/2026/Conference — ICML 2026 regular_

### Official Review · Reviewer_xYqr · 2026-03-03

**Soundness:** 3
**Presentation:** 2
**Significance:** 2
**Originality:** 3
**Overall Recommendation:** 4
**Confidence:** 4

**Summary:**

This paper proposes GiG (Graph-in-Graph), a planning framework for LLM-based embodied agents that addresses context window limitations and hallucinated transitions. GiG encodes environment observations into scene graphs (inner graphs) using a GNN, then organizes those embeddings into state-transition graphs (outer graphs) to track exploration trajectories. GiG is evaluated on Robotouille Synchronous, Asynchronous, and ALFWorld, showing improvements over ReCAP, ReAct, and CoT baselines.

**Compliance With Llm Reviewing Policy:**

Affirmed.

**Final Justification:**

I thank the authors for their response, which resolved most of my concerns.

**Key Questions For Authors:**

1. If the transition function T is known and the environment is fully observable, why not use a classical PDDL planner? What specific advantage does the LLM provide in this setting?
2. How is the experience memory bank constructed? Which model generates the 50 successful trajectories, and what is the computational cost?
3. How sensitive is performance to the size and diversity of the experience memory bank?
4. Can you provide an ablation replacing the GNN encoder with simpler embedding alternatives?

**Limitations:**

yes

**Strengths And Weaknesses:**

# Strengths

1. The paper is generally well-written with a clear logical flow from motivation through method to experiments.

2. The two-tier Graph-in-Graph memory architecture is a creative combination of existing ideas (scene graphs, GAT, trajectory graphs) applied to a relevant problem.

3. The Bounded Lookahead module, while relying on strong assumptions (see weaknesses), provides a clean interface between symbolic transition logic and LLM reasoning by presenting grounded projections rather than asking the LLM to imagine outcomes.

# Weaknesses

1. The BL module relies on a PDDL-derived transition function to project successor states (Eq. 3). This fundamentally limits generality: in ALFWorld where T is unavailable, BL is disabled entirely, meaning one of the main contributions does not apply. More critically, if T is fully known and the environment is fully observable with discrete actions, this is a classical planning problem solvable by PDDL planners. The paper provides no comparison against classical planning baselines, making it unclear why an LLM is needed in the Robotouille setting.

2. The aggregation mechanism for partial observability lacks detail. For ALFWorld, the paper states "we aggregate each step's observations into a single scene graph by merging discovered entities and updating their attributes" but does not describe the actual mechanism. Furthermore, context window efficiency is argued via FLOPs (Fig. 8b) rather than direct measurements (e.g., prompt token counts per step).

3. All benchmarks are text-based symbolic environments, yet the paper claims "embodied" planning. Robotouille and ALFWorld are text-based simulators with discrete actions and structured observations. There is no visual perception, continuous control, or physical interaction. The "embodied" framing overstates the actual evaluation scope. Experiments on environments with richer observation modalities (e.g., visual observations) or real-world settings would strengthen the significance claims.

4. The paper states retrieval "enables cross-task skill transfer" (Sec. 3.3) by using structural similarity without goal filtering. However, all evaluations operate within the same task distribution. A dedicated experiment showing transfer across substantially different task types is needed to support this claim.

5. *Typo in the paper title:* "Lanaguage" → "Language."

6. The notation $\alpha_{u,v}$ in Equation (1) (GAT attention weights) and $\alpha$ in Equation (2) (triplet margin) refer to different concepts. Using distinct symbols would reduce confusion.

---

> ### Author Rebuttal · Authors · 2026-03-30
>
> We appreciate the constructive feedback from the reviewer and address the specific questions below:
>
> **1. LLM vs. PDDL Planner [W1, KQ1]**
>
> The reason we use LLM is because human intents are often under specified in natural language, creating two bottlenecks for PDDL planners that our LLM framework overcomes:
>
> * PDDL planners operate on predicate logic and lack semantic comprehension. A goal like "make a cheeseburger" requires manual translation into formal logic but LLMs can naturally infer these relationships.
>
> * PDDL planners require rigorously defined end-states. A "lettuce, tomato, chicken sandwich on a table" with 3 available tables yields 18 valid configurations (3! ingredient permutations × 3 tables). A PDDL planner requires manually hardcoding all states or writing task-specific programmatic rules. LLMs can skip this by generalizing the goal without enumerating every case.
>
> We want to emphasize that our $T(s,a)$ is compatible with any learned world model, strictly acting as a 1-step dynamics verifier (Sec 3.2) to ground reasoning. Without a forward model, GiG degrades gracefully, achieving 97% on the ALFWorld benchmark.
>
> **2. Aggregation for partial observability, Token Counts & FLOP [W2]**
>
> The aggregation works as a running update of a scene graph. As new observations come in, the system adds objects, updates details, and links them together. For example, it might first detect a “cabinet” and create one node. Later, if it finds a “glass” and a “mug” inside, it adds those as new nodes and connects them to the cabinet. This updated graph is then encoded and treated as the new state, allowing the system to keep a lasting structured memory. A formal algorithm will be provided in the Appendix.
>
> Prior works rely on volatile API costs to show efficiency, we use FLOPs (Figure 8) to provide a standardized, model-agnostic measure of computational cost. We reported per-step reasoning+output token in Figure 9.  We have included per-step prompt token comparison below. GiG reduces the context window consumption.
>
> |Tokens| GiG | ReCAP (window=64) | ReACT |
> | :--- | :--- | :--- | :--- |
> |mean (std)| 12297 (1840) | 40720 (26655) | 57357 (46860) |
>
> **3. Embodied planning claim [W3]**
>
> Modern embodied AI uses a hierarchical paradigm: LLMs handle high-level semantic planning, delegating physical execution to learned policies [1]. Our paper strictly scopes to this high-level planner's memory/reasoning. Using text-based evaluation aligns with established literature (ReCAP, ALFWorld) and isolates planning effectiveness from perception/actuation errors. We will clarify this boundary. To demonstrate robustness, we add an experiment with injected observation noise (adding/removing items). Our experiment results show that GiG remains robust, maintaining stable performance under noise.
>
> | Pass@1 | no-noise | noise/10 steps | noise/5 steps |
> | :--- | :--- | :--- | :--- |
> | GiG  | 93 | 94 | 93 |
> | ReAct | 74 | 67 (-7%) | 62 (-12%) |
> | ReCAP | 71 | 66 (-5%) | 62 (-9%) |
>
> **4. Cross-Task Skill Transfer [W4]**
>
> By "cross-task skill transfer," we mean in-domain compositional transfer. Because GiG retrieves actions via scene graph structure rather than global goals, it can propose a "cutting" sequence from a sandwich task for a burger task. The LLM then acts as a semantic filter, judging whether to adopt or reject these candidate actions. We will update Sec. 3.3 to clarify this mechanism and the distinct challenge of out-of-domain transfer.
>
> **5. Memory bank construction & size analysis & embedding alternative [KQ 2,3,4]**
>
> As noted in Section 4.1, the 50 trajectories were generated by the larger model (Qwen3-235B) across the training seed set. The computational cost for building this static bank is a one-time, negligible upfront cost.
>
> We follow the review's advice by replacing GNN embedding with sentence transformer (all-MiniLM-L6-v2), results show that GNN embedding increases the overall pass@1 rate while reducing the task completion steps compared to sentence transformer.
>
> | Configuration | Score |
> | :--- | :--- |
> | GiG (LD+LA+GNN embedding) | 97 |
> | GiG (LD+LA+Sentence Transformer) | 94 |
>
> | Steps | 0 | 1 | 2 | 3 | 4 | 5 | 6 | 7 | 8 | 9 |
> | :--- | :--- | :--- | :--- | :--- | :--- | :--- | :--- | :--- | :--- | :--- |
> | GNN| 8.9 | 15.6 | 25.5 | 12.1 | 14 | 29.3 | 46.6 | 51.7 | 70.8 | 75.4 |
> | Sentence | 8.9 | 15.8 | 35.7 | 14 | 14.7 | 31.6 | 49.9 | 55.5 | 85.8 | 82.4 |
>
> We sampled 25 traces from the async memory bank across tasks in robotouille async environment and performed the ablation study respect to the memory bank size, the results show as below:
> | # experiences| 50 | 25 | 0 |
> | :--- | :--- | :--- | :--- |
> |Pass@1 | 82 | 78 | 72|
>
> We will correct the typos and notation overlaps in the camera ready version. [W5, W6]
>
> ---
> [1] Ahn et al. (2023). Do As I Can, Not As I Say: Grounding Language in Robotic Affordances, Proceedings of The 6th Conference on Robot Learning, in Proceedings of Machine Learning Research 205:287-318

---

> > ### Author Rebuttal · Reviewer_xYqr · 2026-04-01
> >
> > I appreciate the detailed rebuttal. I consider Q2, Q5, Q6, and Q7 resolved, while Q1, Q3, and Q4 are only partially resolved. My main remaining concerns are the need for stronger justification of the LLM over symbolic planning in the structured benchmark setting; more careful framing of the “embodied” claim; and a more cautious presentation of the cross-task transfer claim unless further evidence is provided.

---

> > > ### Author Response · Authors · 2026-04-03
> > >
> > > We sincerely thank the reviewer for raising their score and helping us refine the precision of our claims. We will integrate the following clarifications into the final version:
> > >
> > > **1. Justification of LLM over Symbolic Planning (Q1)**
> > >
> > > While classical PDDL planners are highly efficient at state-space search, they require exhaustive goal states and cannot process natural language natively. We will clarify in the draft that we use structured benchmarks like Robotouille as a controlled testbed to evaluate the LLM’s semantic planning capability. The LLM naturally translates under-specified human instructions (e.g., "make a burger") into executable actions without requiring a human engineer to manually program every valid end-state configuration, a crucial generalization advantage over classical symbolic methods.
> > >
> > > **2. Framing of the "Embodied" Claim (Q3)**
> > >
> > > We agree on the importance of precise framing. As noted in our rebuttal, modern embodied AI utilizes a hierarchical paradigm where LLMs handle high-level semantic planning while delegating physical execution to learned low-level policies. Our evaluation on text-based simulators (following established literature like ALFWorld and ReCAP) is intentionally designed to isolate and measure this high-level reasoning layer without confounding perception or actuation errors. We will clarify this boundary in the revised manuscript. We will explicitly state in the Introduction that our framework strictly evaluates the abstract, symbolic planning layer of embodied agents rather than the physical embodiment.
> > >
> > > **3. Cautious Presentation of Cross-Task Transfer (Q4)**
> > >
> > > We agree that the phrase "cross-task skill transfer" suggests a broader out-of-domain generalization than we evaluated. We will revise Section 3.3 and adopt the exact terminology from our rebuttal, framing this capability strictly as "in-domain compositional transfer" in the final draft. We will also explicitly note that zero-shot out-of-domain transfer remains an open challenge for future work.

---

### Official Review · Reviewer_bRJg · 2026-03-06

**Soundness:** 3
**Presentation:** 2
**Significance:** 3
**Originality:** 3
**Overall Recommendation:** 4
**Confidence:** 4

**Summary:**

This paper proposes Graph-informed Action Generation (GiG), a hybrid framework for embodied long-horizon task planning that augments LLMs with structured graph memory and bounded symbolic lookahead. The work is motivated by two core limitations of current LLM based embodied planners. (1) the difficulty of maintaining coherent reasoning over long action horizons due to context window constraints, and (2) the tendency of LLMs to hallucinate invalid or constraint-violating actions. To address these issues, the authors introduce a Graph-in-Graph memory structure that combines scene graphs, representing object-level spatial and relational information, with state-transition graphs that encode action-induced dynamics across time. A Graph Neural Network is used to embed these hierarchical graph structures, enabling retrieval of relevant structured priors during planning. These priors guide the LLM’s action generation process. In addition, the framework includes a bounded symbolic lookahead module that performs limited forward simulation to validate or filter candidate actions before execution. The approach is evaluated on Robotouille and ALFWorld, where the authors report substantial improvements in Pass@1 performance over prior LLM-based planning methods.

**Compliance With Llm Reviewing Policy:**

Affirmed.

**Key Questions For Authors:**

It would be great if authors could clarify the following..

1. On ablations, can you more clearly quantify the individual contributions of the GiG retrieval module and the bounded lookahead mechanism? More so, how does performance change when (1) only graph retrieval is used without lookahead, (2) only lookahead is used without structured retrieval, and (3) retrieval depth or clustering granularity is varied?

2. How sensitive is the method to errors in scene graph extraction or partial observability? Have you evaluated performance under injected perception noise or imperfect state representations?

3. How does performance and latency scale as the graph memory grows with longer task horizons or larger environments?

**Limitations:**

While the paper focuses on simulated embodied planning and does not raise immediate high-risk societal concerns, the discussion of limitations and potential broader impacts appears limited. The authors should more explicitly address constraints related to scalability, robustness to perception noise and potential brittleness in real-world robotic deployment. More so, they could discuss risks associated with over-reliance on LLM-guided planning in safety-critical embodied systems, including failure modes where hallucinated but plausible actions bypass safeguards.This clarification would strengthen the paper and would improve the paper's credibility and completeness.

**Strengths And Weaknesses:**

**Soundness**

The approach is conceptually well motivated and technically coherent. The combination of structured graph memory and symbolic validation directly targets known limitations of LLM planners, and empirical results consistently show improvements across benchmarks. However, the evaluation would benefit from stronger ablations isolating the contributions of retrieval and lookahead, clearer reporting of computational cost, and analysis of robustness to noisy or imperfect scene graphs. Overall, the method appears sound but would be strengthened by deeper experimental validation.

**Presentation**

The paper is generally clear and logically structured with a well-articulated motivation and intuitive architectural design. The core ideas are accessible. The results are presented clearly. That said, some implementation details could be explained more thoroughly, and more qualitative examples or failure case analyses would improve clarity and depth. Minor editorial polishing is needed as well.

**Significance**

The work addresses an important and timely challenge in embodied AI of improving reliability in LLM based long horizon planning. By moving beyond prompt-based reasoning toward structured memory and constraint-aware generation, the paper contributes meaningfully to neurosymbolic embodied planning. Its significance is strong within simulation based benchmarks, though real-world robustness remains to be demonstrated.


**Originality**

The paper shows moderate to high originality. While it builds on existing ideas in retrieval-augmented generation, GNNs, and symbolic planning, the hierarchical GiG memory combined with LLM-guided action generation and bounded lookahead represents a novel and coherent integration. The contribution is not merely incremental but a structured advancement in embodied LLM planning.

---

> ### Author Rebuttal · Authors · 2026-03-30
>
> We sincerely thank the reviewer for recognizing the significance, originality of our framework. We appreciate your feedback and address your questions below:
>
> **1. Ablation study of individual module of GiG/retrieval depth/clustering granularity [KQ1]**
>
> We actually provided an ablation study in Section 4.8 (Table 5) to isolate the contribution of individual components. Using only loop detection (GiG+LD) scores 90%, using only bounded lookahead (GiG+BL) scores 80%, and combining them (GiG+LD+BL) yields 93%, adding experience retrieval (GiG+LD+BL+Exp) scores 97%. We followed the reviewer’s suggestion and added the experiment with only experience retrieval module (GiG+Exp). The result shows that with only experience retrieval, GiG achieved 95%.
>
> | | BL | LD | Exp | BL+LD | BL+LD+Exp |
> | :--- | :--- | :--- | :--- | :--- | :--- |
> | Pass@1 | 80 | 90 | 95 | 93 | 97 |
>
> Since the memory retrieval happens at every single step, the retrieval depth (number of actions fetched from the current state) does not affect the performance. As only the actions that are directly related to the current state have more effect. We provide the ablation study on fetched experience depth below:
>
> | | Depth = 1 | Depth = 3 | Depth = 5 |
> | :--- | :--- | :--- | :--- |
> | Pass@1 | 97 | 96 | 97 |
>
> Moreover we would like to clarify that our choice of retrieval threshold ($\tau=0.1$) was not arbitrary, it was derived empirically (Section 4.2). This choice can clearly separate intra-sequence topological steps for efficient retrieval.
>
> **2. Sensitivity to partial observability and perception noise [KQ2]**
>
> GiG is robust to partial observability, which we explicitly evaluated using the ALFWorld benchmark (Section 4.5). In ALFWorld, the environment is partially observable. GiG achieved near-perfect Pass@1 scores (97%) by aggregating partial observations into its dynamic graph memory.
> We agree that perception noise is a critical challenge for physical robotics. To test the model’s planning capability and robustness under perception noise, we perform experiments by directly modifying the text observations to simulate the perception noise. We control the noise level by changing the frequency of noise injection( adding/removing items) during the planning loops (every 5/10 steps). Our experiment results show that GiG remains highly robust, maintaining stable performance under noise.
>
> |Pass@1 | no-noise | noise/10 steps | noise/5 steps |
> | :--- | :--- | :--- | :--- |
> | GiG | 93 | 94 | 93 |
> | ReAct | 74 | 67 (-7%) | 62 (-12%) |
> | ReCAP | 71 | 66 (-5%) | 62 (-9%) |
>
> **3. Scalability of Graph Memory and Latency [KQ3]**
>
> We analyzed scalability in Section 4.7 and Figure 8. As task horizons grow, GiG's computational FLOPs remain orders of magnitude lower than baselines like ReCAP (Figure 8b). Furthermore, the graph construction latency (Figure 8a)  grows linearly with respect to the environment size when tested under a worst-case scenario of rebuilding the entire scene graph from scratch. In practical deployment, the graph is maintained statefully and updated incrementally at each step, resulting in a near-constant, sub-second latency that is negligible compared to LLM decoding times.
>
> **4. Limitations Discussion**
>
> We will expand Section 5 in the camera-ready version to explicitly discuss GiG's unique real-world limitations. Specifically, we will address scalability constraints, noting that infinite-horizon deployments will eventually require an active graph-pruning mechanism as the temporal transition graph grows linearly. We will also discuss the safety-critical risks of deploying LLM-guided planners and how they might perform hallucinated but executable actions (e.g., "wash laptop in sink") that can bypass programmatic safeguards that might cause risks.

---

> > ### Author Rebuttal · Reviewer_bRJg · 2026-04-06
> >
> > This review answers all my questions above. Thanks!

---

> > > ### Author Response · Authors · 2026-04-06
> > >
> > > Thank you for your acknowledgement and for confirming that our rebuttal has fully resolved your concerns. We appreciate your time and the constructive feedback, particularly the push for deeper ablation studies and the noise robustness analysis, which has significantly improved our final manuscript. We are very grateful for your initial positive assessment, where you highlighted the timeliness, originality, and strong empirical results of our framework. Given your confirmation that the remaining weaknesses have been fully addressed, we kindly ask if you might consider raising your score as suggested by the acknowledgement guidelines. Your support is incredibly valuable to us. Please let us know if there is anything else we can provide or clarify in these final days of discussion; we would be more than happy to do so.

---

### Official Review · Reviewer_MdjT · 2026-03-11

**Soundness:** 2
**Presentation:** 3
**Significance:** 2
**Originality:** 2
**Overall Recommendation:** 2
**Confidence:** 4

**Summary:**

**Motivation:**

The motivation of the paper is to resolve how to let VLM to plan in the long-horizon embodied task. The traditional VLM faces the problem including context window shift, hallucinated actions, and cannot handle the tasks in parallel.

**Method:**

To resolve the previous challenges, author proposes three solutions:
1. Graph-in-graph Memory Architecture
2. Experience Retrieval
3. Bounded LookAhead

The author conduct solid experiment in validating the design.

**Compliance With Llm Reviewing Policy:**

Affirmed.

**Final Justification:**

Thank authors for very detailed rebuttal.

This paper tackles an important problem in long-horizon embodied planning with VLMs.

I provide the following reason for rejecting the paper:

a. However, I think for the current version, the author is not highlighting enough the novelty of difference between the paper and existing work in the introduction, I also disagree with the author claiming the major novelty of the paper. thus making the paper more like consolidation of engineering efforts, which is below the bar of ICML, which is a top ML conference.

b. The Bounded LookAhead relies on access to a transition function T(s, a), which is difficult to obtain in physical environments, which is mostly the reason why I think a real-robot experiment would be very important, the author fails to conduct, and fails to give any explanations regarding the robustness of the system. The physcial robot experiment, as a case study, would be vital to test the model's robustness and scalability.

c. In the meantime, the ablation studies of memory retrieval should be included in the main paper. I also read other reviewer's comments, and think the current experiments results are not solid enough to support the paper's claim, including clustering granularity, retrieval depth experiments is not conducted with any detailed ablation setups.

It could be a paper with more clear writings and improved experiments, I encourage authors for refining the updated version, for now I think it is below the acceptance bar, so I confirm my rating with a reject. Thanks authors again for their hard works in rebuttal.

**Key Questions For Authors:**

1. More clear statement over the novelty based on GraphRAG、Graph-of-Thought、ReCAP, more clear statement regarding the advantage (especially experiment analysis) using Graph-in-Graph rather than single graph.
2. Please see weakness, the Bounded Lookahead requires transition functions, is the methodology still work when there are no clear transition functions, especially in real-world set up?
3. Comparisons Memory Retrieval to top-k retrieval or goal-aware retrieval.
4. Additional experiments on real-robot setup.

**Limitations:**

Yes, the author clearly states the limitations of their work.

**Strengths And Weaknesses:**

### Strength:

1. The problem formulation is clear, context drift, hallucinated transitions and sequential planning limitation are challenging and important quesetions.
2. Graph-based memory is novel.
3. The experiment results are solid and even small model achieves about 15% performance improvement.

### Weakness:

1. The novelty of the paper is limited, scene graph, node is already existed in other related works, the paper looks more like an enginneeringly solid paper other than theory-level breakthrough.
2. Bounded LookAhead requires transition function T(s,a), which is hard to acquire in the simulator. Also, BL can not be used in ALFWorld.
3. Only simulation experiment, no real-robot setup.

---

> ### Author Rebuttal · Authors · 2026-03-30
>
> We appreciate the reviewer recognizing the novelty of our Graph-in-Graph memory architecture. We address your insightful questions below:
>
> **1. Clarification of Novelty [W1,KQ1]**
>
> We argue that GiG is not just an engineering optimization, but rather a novel memory architecture for embodied planning. We highlight the differences between GiG and existing literature below:
>
> * Standard GraphRAG: Operates on a knowledge graph with semantic edges, missing both temporal evolution (GiG's outer graph) and spatial structure (GiG's inner graph). GiG dynamically builds a topological representation of the environment on the fly, transforming successful transient exploration steps into a persistent, retrievable memory to guide future actions.
>
> * GoT: Structures internal LLM "thoughts." In contrast, GiG nodes represent concrete scene realities. The inner graph represents the explicit spatial topology of the environment, and the outer graph represents the temporal history of states the agent has visited and the physical actions taken. This is entirely distinct from mapping an LLM’s internal thought process.
>
> * ReCAP: Maintains a hierarchical context tree to recursively decompose high-level goals into sub-tasks. Its major limitation is "artificial serialization": in a tree topology, sibling nodes (parallel sub-tasks) are structurally blocked until the current node completes (Figure 1). GiG models state as a dynamic graph rather than a rigid top-down tree, allowing instantiate new sub-goals based on real-time feasibility, natively supporting interleaved actions.
>
> * Single Graph: A single graph would confound spatial topology with temporal evolution. By decoupling them, the inner scene graph explicitly captures instantaneous physical relations, while the outer transition graph explicitly tracks task progression and detects cyclic failures.
>
> **2. Bounded Lookahead (BL) and Transition Functions in Real-World Setups [W2,KQ2]**
>
> While in the Robotouille experiment we utilized a simulator-based forward model $T$, we emphasize that $T$ does not perform search or solve the task. The BL module uses $T$ strictly as a 1-step dynamics verifier (Section 3.2) to ground the reasoning. For example, given the action 'put an apple on table', $T$ simply returns the deterministic outcome: 'hand empty, apple on table'. The LLM remains responsible for all semantic reasoning and planning. Furthermore, the BL module acts as an interface compatible with any learned real-world transition model. As the reviewer identified, even in ALFWorld, where $T$ is not available due to partial observability, GiG still guarantees graceful degradation. The core Graph-in-Graph architecture achieves 97% Pass@1 on Qwen3 and DeepSeek, outperforming baselines.
>
> **3. Memory Retrieval: K=1 vs. Top-K [KQ3]**
>
> Top-K retrieval often floods the context window with trajectories from neighboring states that conflict with the agent's current reality. Because our GNN effectively separates consecutive states (Section 4.2), K=1 enforces a precise, step-by-step receding horizon. As shown in our ablation study below, increasing K yields no benefit and slightly degrades Pass@1 performance due to context confusion.
> || K=1 (Default) | K=3 | K=5 |
> | :--- | :--- | :--- | :--- |
> | **Pass@1** | 97 | 97 | 95 |
>
> **4. Real-Robot Experiments [W3,KQ4]**
>
> We agree that physical deployment is the ultimate goal. However, modern embodied AI relies on a hierarchical paradigm: utilizing LLMs for high-level semantic planning and task decomposition, while delegating physical execution to low-level learned policies [1]. Our paper is explicitly scoped to advance the first component. In line with literature (e.g., ReCAP), we evaluate community-standard benchmarks (Robotouille, ALFWorld) to rigorously measure the planning capability of our agent. Evaluating our planner directly on real robots would introduce noise from low-level execution and vision systems, acting as confounding variables that obscure the core algorithm's effectiveness.
>
> To bridge the gap and address the concern about robustness, we conducted a new study by introducing simulated perception noise (adding/removing items from raw observation text). The results demonstrate that GiG’s Graph-in-Graph memory remains highly robust, maintaining stable performance under noise. Integrating GiG with a low-level physical controller remains an exciting next step for future work.
>
> | | noise-level | no-noise | noise/10 steps | noise/5 steps |
> | :--- | :--- | :--- | :--- | :--- |
> | GiG | Pass@1 | 93 | 94 | 93 |
> | ReAct | Pass@1 | 74 | 67 (-7%) | 62 (-12%) |
> | ReCAP | Pass@1 | 71 | 66 (-5%) | 62 (-9%) |
>
> ---
>
> [1] Ahn et al. (2023). Do As I Can, Not As I Say: Grounding Language in Robotic Affordances, Proceedings of The 6th Conference on Robot Learning, in Proceedings of Machine Learning Research 205:287-318

---

> > ### Author Rebuttal · Reviewer_MdjT · 2026-04-02
> >
> > Thanks author for very detailed rebuttal.
> >
> > a. However, I think for the current version, the author is not highlighting enough the novelty of difference between the paper and existing work in the introduction, thus making the paper more like consolidation of engineering efforts, which is below the bar of ICML, which is a top ML conference.
> >
> > b. The Bounded LookAhead relies on access to a transition function T(s, a), which is difficult to obtain in physical environments, which is mostly the reason why I think a real-robot experiment would be very important. "Modern embodied AI relies on a hierarchical paradigm: utilizing LLMs for high-level semantic planning and task decomposition, while delegating physical execution to low-level learned policies", doing a physical robot in the real robot setup using existing policy baselines like hueristic pick and place would be grealy to resolve the concern, instead of only testing the method in simulation, which is not convising that whether the method has the true potential for deployment, expecially consider how to acquire transition functions in physical robots. Many existing related works which focusing on embodied planning like [1][2] always combine with a solid physical experiments and demos, these are not execuses because you focus on "embodied domain", and using heuristic methods (like off-the-shelf pick and place) as low-level policy is not considered as an extention of the work,  should be easy to conduct, and is a very important experiment results of this work, however authors fail to provide.
> >
> > [1] MAGIC, arXiv:2411.09627
> > [2] BLADE, Learning Compositional Behaviors from Demonstration and Language, CoRL 2024

---

> > > ### Author Response · Authors · 2026-04-02
> > >
> > > We thank the reviewer for continued engagement and feedback. However, the rebuttal feedback does not provide any new evidence or detailed justification of previous review and we address the concerns below:
> > >
> > > **Novelty concern**
> > >
> > > In Section 2 (Related Work) of the paper and Rebuttal Section 1, we provided a comprehensive and rigorous distinction between GiG and existing memory structures (GraphRAG, GoT, ReCAP). We would like to note that neither the original review nor the rebuttal feedback provides any specific details regarding the lack of comparison with existing work, leaving us unable to address this point further.
> > >
> > > **Real world setup and transition function concern**
> > >
> > > We appreciate the reviewer highlighting works like MAGIC[1] and BLADE[2], which illustrate the complexities of physical deployment. However, we respectfully note a fundamental difference in scope between MAGIC[1], BLADE[2],  and GiG. Neither MAGIC nor BLADE use LLM as their core reasoning engine for dynamic task planning. BLADE focuses on advancing low-level continuous control, relying on a classical symbolic search algorithm (Fast-Forward [3]) rather than an LLM for its abstract planning. Also, as noted in BLADE, directly learning a single goal-conditioned policy that can generalize to novel states and goals is challenging. To overcome this, they do not rely on simple "off-the-shelf" heuristics; rather, they dedicate massive engineering efforts to advancing low-level continuous control (i.e. MAGIC's "two-stage contact-point matching process" or BLADE's requirement to collect human data for "training visuomotor policies for individual skills"). Because their primary contributions lie in these physical execution layers, physical hardware demonstrations are essential to validate their work.
> > >
> > > Our core novelty resides exclusively at the LLM memory and reasoning level for long-horizon planning, rather than low-level continuous control. Because we are explicitly scoped to advance the high-level semantic planner, we evaluated on simulated benchmarks to mathematically isolate our contribution in structured LLM memory and planning from the unmodeled execution failures that physical controllers would inevitably introduce. In the LLM planning domain, most papers[4][5] use standard benchmarks such as Robotouille and ALFWorld which allow for a fair, large-scale comparison against SOTA baselines that would be impossible to replicate on a single, custom real-robot setup.
> > >
> > > Furthermore, regarding the transition function T(s,a), we are aware that a perfect forward model is difficult to obtain in physical environments. This is precisely the reason adding our GiG framework improves planning robustness for an LLM-agent. The ALFWorld experiments were conducted specifically because the environment lacks an explicit transition function due to partial observability. In this setting, GiG bypasses the Bounded Lookahead module entirely and relies solely on the core Graph-in-Graph memory fallback, yet still achieves state-of-the-art performance (97% Pass@1).
> > >
> > > While physical hardware integration falls outside the scope of our research focus on LLM memory, we agree it is a critical next step for the broader embodied AI community. In our revised manuscript, we will expand our discussion to detail how GiG’s robust fallback mechanisms lay the groundwork for researchers to integrate our planner with downstream physical controllers.
> > >
> > > ---
> > >
> > > [1] MAGIC, arXiv:2411.09627
> > >
> > > [2] BLADE, Learning Compositional Behaviors from Demonstration and Language, CoRL 2024
> > >
> > > [3] Hoffmann and B. Nebel. The FF planning system: Fast plan generation through heuristic search. JAIR, 14:253–302, 2001. 5
> > >
> > > [4] Zhang, Z., Chen, T., Xu, W., Pentland, A., and Pei, J. Recap: Recursive context-aware reasoning and planning for large language model agents. In Advances in Neural Information Processing Systems, NeurIPS, 2025
> > >
> > > [5] Yao, S., Zhao, J., Yu, D., Du, N., Shafran, I., Narasimhan, K., and Cao, Y. ReAct: Synergizing reasoning and acting in language models. In International Conference on Learning Representations, ICLR, 2023

---

### Official Review · Reviewer_P3PY · 2026-03-13

**Soundness:** 2
**Presentation:** 3
**Significance:** 2
**Originality:** 3
**Overall Recommendation:** 4
**Confidence:** 2

**Summary:**

The paper introduces 'Graph-in-Graph' memory architecture, where GNN encodes scene graphs embeddings to capture dynamic environmental states and use experience retrieval for better task planning.

The paper proposed 'Bounded Lookahead' to enable proactive optimization by grounding the agent's reasoning with environment constraints.

The experiments on three benchmarks, showing the proposed method outperforms state-of-art baselines with efficient computational costs.

**Compliance With Llm Reviewing Policy:**

Affirmed.

**Final Justification:**

My concerns have been adequately addressed. I keep my score as a positive assessment.

**Key Questions For Authors:**

1. Regarding the model size in the Limitation, how does performance scale with model size when memory is added and can the proposed approach narrow the gap between smaller models and 100B+ models?

2. What is the end-to-end inference latency during deployment and is the method practical for real-time embodied planning under realistic hardware constraints?

3. How sensitive is performance to reasoning chain length?

**Limitations:**

yes

**Strengths And Weaknesses:**

Strengths:

1. the paper is well-written, easy to follow, presentations is good
2. the proposed methods are reasonable to tackle the addressed challenges
3. experiment results are strong.

Weaknesses:

Overall, the method appears to work well within the benchmarks, but I'm not yet convinced that is is a compelling approach for general robotics/embodied applications. With more complex environments, the graph size will be increased and structure will be more complicated, then handling the search space would be a bottleneck. Also the approach is relying on LLMs (as mentioned in Limitation). However I would be open to revising review if the authors can address these questions.

---

> ### Author Rebuttal · Authors · 2026-03-30
>
> We thank the reviewer for recognizing the strong experiment results and our novel graph-in-graph memory architecture. We address the weaknesses and questions below:
>
> **1. Scalability to Complex Environments and Search Space Bottlenecks [W]**
>
> We agree that scalability is critical for efficient memory retrieval. However, GiG avoids search bottlenecks through two design choices:
> *  Context Management: We retrieve only the single most relevant immediate transition (Section 3.3) based on scene graph embedding rather than matching full-history, making the context window manageable and search efficient.
> *  Vector Similarity Search: As detailed in Section 4.1, we utilize the Faiss library for vector similarity search. Since the GNN condenses the scene graph into a dense, fixed-size vector $\mathbf{z}_t$, similarity matching remains highly scalable and operates in sub-millisecond time frames, even as the experience bank grows [1][2].
>
> We agree that embodied applications may run into problems such as perception noise, actuation errors. However, modern embodied AI relies on a hierarchical paradigm: utilizing LLMs for high-level semantic planning and task decomposition, while delegating physical execution to low-level learned policies [3]. Our paper is explicitly scoped to advance the first component. To show the robustness of our framework, we performed a robustness study. We systematically add/remove items from the raw observation text at different frequencies to simulate perception noise, and feed the noisy text to GiG. The results demonstrate that GiG’s Graph-in-Graph memory remains highly robust, maintaining stable performance under noise, while baseline methods degrade.
>
> | Pass@1 | no-noise | noise/10 steps | noise/5 steps |
> | :--- | :--- | :--- | :--- |
> | GiG | 93 | 94 | 93 |
> | ReAct | 74 | 67 (-7%) | 62 (-12%) |
> | ReCAP  | 71 | 66 (-5%) | 62 (-9%) |
>
>
> **2. Model Size Scaling and Narrowing the Performance Gap [KQ1]**
>
> We are glad the reviewer identifies a key contribution where our proposed method excels. In our experiments, we observe that the added experience memory narrows the capability gap between smaller models (Qwen3-30B and Gemini-2.5-Flash-Lite) and their larger counterparts (Qwen3-235B and Gemini-2.5-Flash).
> As demonstrated in Section 4.6 (Table 4), introducing the experience memory (GiG+Exp):
> * Improved the Pass@1 rate of the Qwen3-30B by an absolute **15%** (from 27% to 42%).
> * Improved the Gemini-2.5-Flash-Lite by **7%** (from 19% to 26%).
>
> While it does not fully close the gap to the larger models, it proves that smaller models can effectively leverage structural experience to achieve better performance.
>
> **3. End-to-End Inference Latency and Real-Time Practicality [KQ2]**
>
> As analyzed in Section 4.7 and Figure 8(a), the overhead introduced by our method (milliseconds) is negligible compared to the LLM reasoning latency (seconds). We also want to clarify a detail not explicitly highlighted in the main text: the near-linear graph build latency shown in Figure 8(a) represents a worst-case upper bound. In practical deployment, the scene graph can be maintained statefully and updated incrementally, modifying only the few nodes or edges that change per interaction, making the graph overhead negligible. However, we agree that the practicality of "real-time" deployment depends entirely on the task frequency. Because the primary bottleneck remains the LLM inference latency, our method is highly practical for low-frequency, high-level semantic planning (e.g., task decomposition, navigating environments, operating appliances) where decision horizons naturally operate on the order of seconds. It is not suitable for high-frequency reactive control. We will make this clarification in the limitation section.
>
> **4. Sensitivity to Reasoning Chain Length [KQ3]**
>
> Our ablation study in Section 4.8 (Figure 9) directly investigated token generation lengths (which include both reasoning + output, since the output is just a one sentence instruction, the tokens count here can reflect the reasoning length). We do not restrict or control the reasoning chain length. We found that the length of the reasoning chain is an indicator of task difficulty rather than an independent variable we tune. The models consistently produced longer reasoning traces (generating more tokens) on failed tasks compared to successful ones, as the agent intensifies its "mental effort" when stuck.
>
> ---
> [1] Douze, M., Guzhva, A., Deng, C., Johnson, J., Szilvasy, G., Mazaré, P. E., ... & Jégou, H. (2025). The faiss library. IEEE Transactions on Big Data.
>
> [2] Johnson, J., Douze, M., & Jégou, H. (2019). Billion-scale similarity search with GPUs. IEEE transactions on big data, 7(3), 535-547.
>
> [3] Ahn et al. (2023). Do As I Can, Not As I Say: Grounding Language in Robotic Affordances, Proceedings of The 6th Conference on Robot Learning, in Proceedings of Machine Learning Research 205:287-318

---

> > ### Author Rebuttal · Reviewer_P3PY · 2026-04-06
> >
> > I want to thank authors for detailed responses. My concerns have been adequately addressed. I keep my score as a positive assessment.

---

### Decision · Program_Chairs · 2026-04-30

**Decision:**

Accept (regular)

**Comment:**

This paper presents Graph-in-Graph (GiG), a hierarchical memory architecture designed to improve long-horizon task planning for LLM-based agents. The primary contribution—decoupling spatial scene graphs from temporal transition graphs—is technically sound and addresses critical issues of context drift and hallucinated transitions in embodied planning. Three reviewers recommend a weak accept, praising the framework’s originality and strong empirical performance on benchmarks like Robotouille and ALFWorld. One reviewer maintained a reject, citing limited novelty and the lack of real-robot experiments. However, after carefully reading the reviews and the extensive rebuttal, the Area Chair finds that the authors have successfully addressed the majority of concerns. Specifically, the authors provided new ablation studies on noise robustness and retrieval depth that satisfied the other three reviewers. Regarding the reject, the reviewer’s insistence on physical hardware demos appears to be a scope mismatch; the paper is explicitly focused on high-level LLM memory architecture rather than low-level continuous control. Furthermore, the reviewer’s final justification contained factual inaccuracies—such as misidentifying the LLM-based method as a VLM—which suggests a less-than-thorough reading of the core methodology. The authors have committed to refining the "embodied" framing and toning down cross-task transfer claims to better reflect the simulation-based evaluation. Given the technical coherence of the GiG framework and its clear benefits for smaller models, the paper is a solid contribution to the field of neurosymbolic planning.